# Brain insulin resistance impairs hippocampal synaptic plasticity and memory by increasing GluA1 palmitoylation through FoxO3a

Matteo Spinelli[1], Salvatore Fusco[1,2], Marco Mainardi [1], Federico Scala[1], Francesca Natale [1], Rosita Lapenta[3] Andrea Mattera[1], Marco Rinaudo [1], Domenica Donatella Li Puma [1], Cristian Ripoli [1], Alfonso Grassi[3], Marcello D'Ascenzo [1] & Claudio Grassi [1,4]

High-fat diet (HFD) and metabolic diseases cause detrimental effects on hippocampal synaptic plasticity, learning, and memory through molecular mechanisms still poorly understood. Here, we demonstrate that HFD increases palmitic acid deposition in the hippocampus and induces hippocampal insulin resistance leading to FoxO3a-mediated over-expression of the palmitoyltransferase zDHHC3. The excess of palmitic acid along with higher zDHHC3 levels causes hyper-palmitoylation of AMPA glutamate receptor subunit GluA1, hindering its activity-dependent trafficking to the plasma membrane. Accordingly, AMPAR current amplitudes and, more importantly, their potentiation underlying synaptic plasticity were inhibited, as well as hippocampal-dependent memory. Hippocampus-specific silencing of Zdhhc3 and, interestingly enough, intranasal injection of the palmitoyltransferase inhibitor, 2-bromopalmitate, counteract GluA1 hyper-palmitoylation and restore synaptic plasticity and memory in HFD mice. Our data reveal a key role of FoxO3a/Zdhhc3/GluA1 axis in the HFD-dependent impairment of cognitive function and identify a novel mechanism underlying the cross talk between metabolic and cognitive disorders.

[1] Institute of Human Physiology, Università Cattolica Medical School, 00168 Rome, Italy. [2] San Raffaele Pisana Scientific Institute for Research, Hospitalization and Health Care, 00163 Rome, Italy. [3] Department of Chemistry and Biology, University of Salerno, 84084 Salerno, Italy. [4] Fondazione Policlinico Gemelli, 00168 Rome, Italy. Correspondence and requests for materials should be addressed to S.F. (email: salvatore.fusco@unicatt.it)

Hippocampal synaptic plasticity plays a central role in cognitive function[1]. During learning and memory, activity-dependent functional plasticity causes structural changes that are essential for the acquisition of new information[2]. This is well exemplified by the long-term potentiation (LTP) paradigm, a cellular correlate of learning and memory[3], in which glutamate released following high-frequency stimulation of pre-synaptic terminals induces N-methyl-D-aspartate (NMDA) receptor/CaMKII signaling activation and recruitment of α-amino-3-hydroxy-5-methyl-4-isoxazolepropionic acid (AMPA) receptors at the postsynaptic site, thereby enhancing the amplitude of excitatory postsynaptic currents (EPSCs)[4].

Post-translational modifications have emerged as critical regulators of synaptic transmission and plasticity[5, 6]. In particular, phosphorylation and palmitoylation of both NMDA and AMPA receptor (NMDAR and AMPAR) subunits control stability, trafficking, protein–protein interaction, and synaptic expression of glutamate receptors (GluRs) in the central nervous system[7–9]. Phosphorylation and palmitoylation are labile and reversible modifications that can be dynamically controlled by extracellular and environmental stimuli[10, 11].

Recently, emerging attention has been devoted to the impact of nutrients and diet on neuronal network development and activity[12]. Experimental models of overnutrition and metabolic diseases (e.g., obesity and insulin resistance) show severe learning and memory defects[13]. High-fat diet (HFD) is the most commonly used experimental model of metabolic disease, causing both peripheral insulin resistance and detrimental effects on brain function[14], but the molecular mechanisms underlying the impact of nutrient excess on cognitive function are still poorly understood.

Palmitic acid is the most abundant fatty acid in the brain and, importantly, palmitoylation consists of a covalent attachment of a palmitate molecule to proteins[15]. Protein palmitoylation is finely regulated by a class of enzymes, the protein acyl transferases (PATs) containing an aspartate-histidine-histidine-cysteine (DHHC) domain[16]. However, so far no information is available on whether: (i) HFD affects synaptic protein palmitoylation and (ii) this molecular mechanism underlies cognitive function alterations associated with brain insulin resistance.

Here, we demonstrate that HFD-induced brain insulin resistance causes LTP and memory impairment due to the accumulation of palmitic acid and increased expression/activation of zDHHC3 leading to hyper-palmitoylation of GluA1 in the hippocampus. In vitro stimulation of hippocampal neurons with both insulin and palmitic acid reproduces the in vivo molecular changes, affects the recruitment of GluA1 to the synaptic membrane, and inhibits AMPA currents at glutamatergic synapses under both basal conditions and following LTP protocols. Moreover, hippocampus-selective silencing of zDHHC3 or overexpression of the palmitoylation-deficient GluA1 mutant rescue the synaptic plasticity deterioration induced by insulin resistance. Finally, mice treatment with the palmitoylation inhibitor 2-bromopalmitate (2-BP) abolishes the detrimental effects of HFD on learning and memory. These data suggest that aberrant GluA1 palmitoylation plays a critical role in hippocampal synaptic plasticity impairment and cognitive decline observed in experimental models of metabolic diseases.

## Results

### HFD induces brain insulin resistance and LTP impairment.
Epidemiological and experimental evidence indicate that HFD, in addition to causing peripheral metabolic changes including insulin resistance and fatty acid deposition, impairs hippocampal plasticity[17, 18]. To investigate the mechanism underlying the impairment of hippocampal synaptic plasticity in HFD mice and to determine the role of hippocampal insulin signaling in these alterations, we performed electrophysiological, behavioral, and metabolic analyses in C57BL/6 mice after 6 weeks of HFD or standard diet (SD).

In a first cohort of mice, we found that LTP induced at the CA3-CA1 hippocampal synapses by high-frequency stimulation (HFS) was significantly reduced in slices from HFD mice ($33.5 \pm 6.4\%$ vs. $81.3 \pm 6.6\%$; Fig. 1a). Accordingly, HFD impaired hippocampus-dependent learning and memory assessed by the novel object recognition (NOR) and Morris water maze (MWM) tests. HFD mice showed less preference for the novel object than controls (Supplementary Fig. 1a). Moreover, HFD significantly increased the latency to find the hidden platform during the training of MWM and reduced the time spent in the target quadrant during the probe test (Supplementary Fig. 1b,c). Next, we evaluated hippocampal fatty acid concentrations and the insulin plasma levels in a second cohort of mice. Hippocampi of HFD mice showed higher contents of palmitic ($+111 \pm 8\%$), stearic ($+128 \pm 7\%$), and oleic ($+94\% \pm 12\%$) acids than controls (Fig. 1b). Moreover, higher plasma levels of insulin were found in HFD mice ($4.42 \pm 0.13$ ng mL$^{-1}$ vs. $2.93 \pm 0.08$ ng mL$^{-1}$; Fig. 1c), in accordance with their peripheral insulin resistance. To test the hippocampal insulin sensitivity of HFD mice, we assessed phosphorylation of both Akt and its main downstream effectors, GSK3β and FoxO3a, 30 min after intranasal injection of insulin. We observed increased phosphorylation levels of Akt, GSK3β, and FoxO3a following insulin injection in controls (Fig. 1d,e). Conversely, in the hippocampi of HFD mice aberrant insulin signaling was found, consisting of basal hyper-phosphorylation of both Akt and GSK3β kinases (pAkt Ser$^{473}$: $+ 210 \pm 23\%$; pGSK3β Ser$^9$: $+ 251 \pm 31\%$) and loss of insulin-dependent phosphorylation of Akt, GSK3β, and FoxO3a (pFoxO3a Ser$^{253}$) ($F_{3.28} = 52.35$ for AKT Ser$^{473}$, SD$_{veh}$ vs. SD$_{ins}$ $p = 0.00039$, SD$_{veh}$ vs. HFD$_{veh}$ $p = 0.00019$; $F_{3.28} = 36.03$ for GSK3β Ser$^9$, SD$_{veh}$ vs. SD$_{ins}$ $p = 0.00011$, SD$_{veh}$ vs. HFD$_{veh}$ $p = 0.00096$; $F_{3.28} = 53.72$ for pFoxO3a Ser$^{253}$, SD$_{veh}$ vs. SD$_{ins}$ $p = 0.0011$; Fig. 1d,e). These data indicated that impaired synaptic plasticity and memory were associated with altered insulin signaling and increased fatty acid deposition in the hippocampi of HFD mice.

### HFD increases GluA1 palmitoylation and zDHHC3 expression.
Many studies have shown that post-translational modifications of GluR subunits, such as palmitoylation and phosphorylation, play critical roles in the regulation of synaptic plasticity[19–21]. An intriguing hypothesis was that an excess of metabolic factors, such as insulin and palmitic acid, could impinge on palmitoylation of GluRs. Therefore, we first analyzed the palmitoylation of NMDA and AMPA glutamate receptor subunits in the hippocampi of SD and HFD mice by acyl-biotin exchange (ABE) assay. HFD mice exhibited increased palmitoylation of both GluA1 and GluA2 subunits of AMPAR ($+ 40 \pm 6.1\%$ and $+ 18 \pm 3.5\%$, respectively, vs. SD; Fig. 2a, b). Conversely, no changes in palmitoylation levels of NMDA or other glutamate receptor subunits, as well as of PSD95, were observed (Supplementary Fig. 2a, b). Accordingly, GluA1 phosphorylation at serine 845 (pGluA1 Ser$^{845}$), an activation site negatively regulated by palmitoylation, was significantly inhibited in the hippocampi of HFD mice ($-38 \pm 3.7\%$ vs. SD; Fig. 2c, d). Instead, GluA2 phosphorylation at serine 880 (pGluA2 Ser$^{880}$) was not affected by the dietary regimen (Fig. 2c, d).

Protein palmitoylation is primarily regulated by zinc finger DHHC-type palmitoyl transferases (ZDHHCs)[22]. We examined the expression of the main zDHHCs (2, 3, 4, 5, 7, 8, 12, 13, 15, 17, 20) triggering palmitoylation of synaptic proteins including both

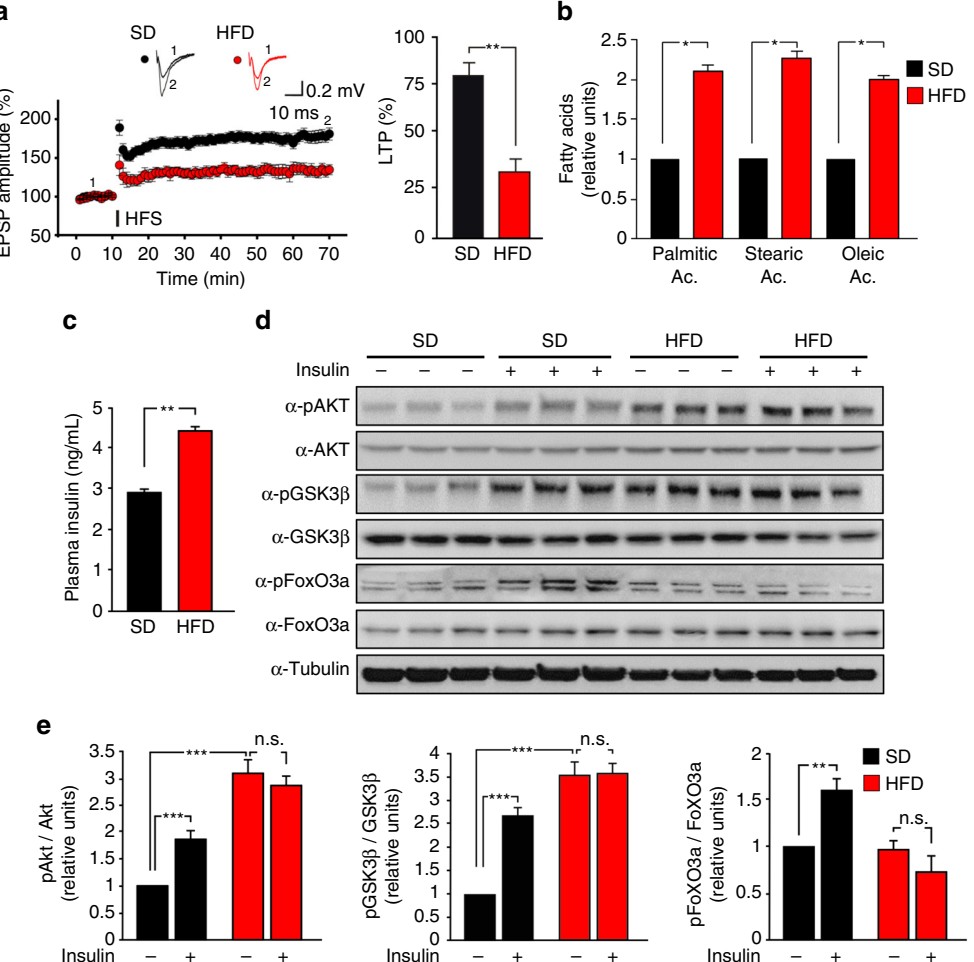

**Fig. 1** HFD impairs synaptic plasticity, induces insulin resistance, and increases palmitic acid levels in the hippocampus. **a** Time course of LTP at CA3-CA1 synapses induced by HFS delivered at time 10 (line) in hippocampal slices obtained from mice fed with SD ($n = 12$ slices) or HFD ($n = 9$ slices) for 6 weeks. Results are expressed as percentages of baseline fEPSP amplitude (=100%). Insets show representative fEPSPs at baseline (1) and during the last 5 min of LTP recording (2). Traces are averages of five consecutive responses at the time points indicated with 1 and 2. On right, bar graphs of LTP observed during the last 5 min in SD and HFD mice (statistics by unpaired Student's *t*-test). **b** Relative amounts of fatty acid (measured by GC-FID analysis) extracted from hippocampi of SD ($n = 9$) and HFD ($n = 8$) mice (statistics by unpaired Student's *t*-test). **c** Insulin plasma levels of SD and HFD mice measured by ELISA performed in duplicate ($n = 10$ mice per group; statistics by unpaired Student's *t*-test). **d** Immunoblot analysis revealing increased phosphorylation of Akt Ser[473] and GSK3β Ser[9] and abolished responsiveness to insulin injection in the hippocampi of HFD mice. Samples were harvested from two independent experiments. **e** Densitometry of phospho-proteins (shown in **d**) normalized to both the corresponding total protein and tubulin ($n = 6$ per group; statistics by two-way ANOVA and Bonferroni post hoc). Data are expressed as mean ± standard error of the mean (SEM). *$p < 0.05$; ** $p < 0.01$; ***$p < 0.001$; n.s. not significant. See also Supplementary Fig. 1

AMPAR and NMDAR subunits. zDHHC3 was the only PAT transcriptionally upregulated in the hippocampi of HFD mice (+ 196 ± 22%; Fig. 2e). Moreover, the activity of zDHHC3 is finely regulated by its autopalmitoylation[23]. In the hippocampus of HFD mice we found increased levels of palmitoylated zDHHC3 (Fig. 2f). Collectively, our ex vivo data suggested that HFD heightened GluA1 palmitoylation through a dual mechanism: (i) increased availability of the substrate, i.e., the palmitic acid and (ii) enhanced expression and palmitoylation of the palmitoyl-transferase zDHHC3.

**IPA transcriptionally enhances GluA1 palmitoylation.** To identify the molecular mechanism underlying the HFD-induced GluA1 hyper-palmitoylation, we set up an in vitro model of neuronal insulin resistance resembling the metabolic and molecular changes observed in vivo. Based on our data suggesting that both insulin and palmitic acid are critical for the development of insulin resistance, we cultivated hippocampal neurons for 24 h

with either insulin (20 nM) alone or a cocktail of both insulin and palmitic acid (20 nM and 0.2 mM, respectively, hereafter named IPA) and analyzed the insulin signaling. Both protocols of insulin resistance abolished insulin-mediated phosphorylation of Akt and GSK3β, but only IPA treatment induced the inhibitory hyper-phosphorylation of GSK3β at Ser[9] (+139 ± 10%; Fig. 3a, b). Interestingly, IPA was also the only protocol able to downregulate the inhibitory phosphorylation of FoxO3a (−70 ± 12%; Fig. 3a, b), leading to the hyper-activation of the transcription factor ($F_{3.32} = 35.55$ for AKT Ser[473], $CTR_{veh}$ vs. $CTR_{ins}$ $p = 0.002$, $CTR_{veh}$ vs. $INS_{veh}$ $p = 0.02$, $CTR_{veh}$ vs. $IPA_{veh}$ $p = 0.018$; $F_{3.32} = 57.88$ for GSK3β Ser[9], $CTR_{veh}$ vs. $CTR_{ins}$ $p = 0.009$, $CTR_{veh}$ vs. $IPA_{veh}$ $p = 0.0003$; and $F_{3.32} = 11.63$ for pFoxO3a Ser[253], $CTR_{veh}$ vs. $IPA_{veh}$ $p = 0.006$). More importantly, chronic stimulation with IPA, but not insulin alone, enhanced the expression of zDHHC3 at the protein level (+190 ± 20%; $F_{5.14} = 43.16$, CTR vs. IPA $p = 0.013$; Fig. 3c).

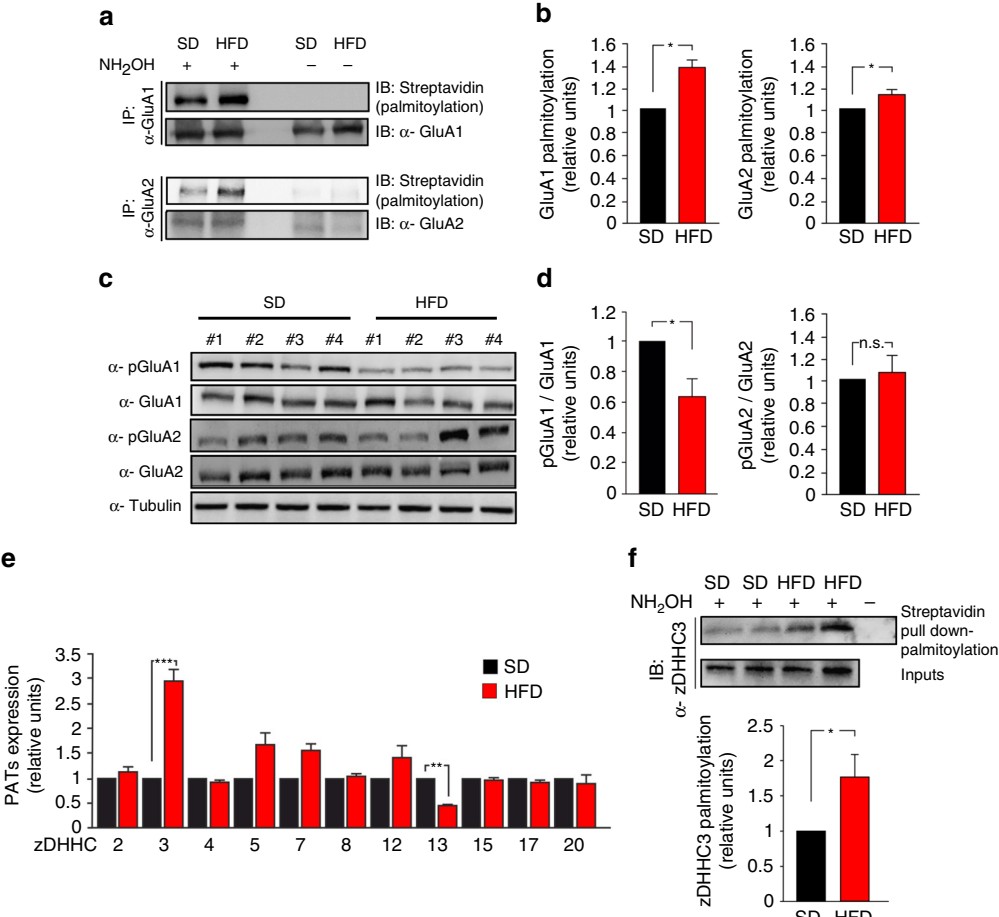

**Fig. 2** HFD increases palmitoylation and inhibits phosphorylation of GluA1. **a** Palmitoylation of GluA1 and GluA2 was examined in the hippocampus of SD and HFD mice using a modified biotin switch assay (ABE, see "Acyl-biotinyl exchange assay" section in Methods). Immunoblot showing palmitoylated (acyl-biotinyl exchanged and detected by streptavidin) GluR (top) and total immunoprecipitated protein (bottom). Samples without $NH_2OH$ are negative controls. **b** Densitometry of palmitoylated GluA1/total immunoprecipitated GluA1 (left, $n = 6$) and palmitoylated GluA2/total immunoprecipitated GluA2 ratio (right, $n = 4$; statistics by Mann–Whitney test). **c** Immunoblots of hippocampal homogenates revealing reduced phosphorylation of GluA1 at serine 845 (pGluA1 Ser[845]) in HFD mice, and unchanged phosphorylation of GluA2 at serine 880 (pGluA2 Ser[880]). Samples were harvested from two independent experiments. **d** Densitometry of pGluA1 Ser[845] (left) and pGluA2 Ser[880] (right) blots normalized to both the corresponding total protein and tubulin ($n = 10$ mice per group; statistics by unpaired Student's $t$-test). **e** Expression of zDHHC 2, 3, 4, 5, 7, 8, 12, 13, 15, 17, and 20 mRNA, assessed by Real-Time qPCR. Gene expression was normalized to actin. Data represent mean values obtained from five mice for each group; experiments were performed in triplicate (statistics by unpaired Student's $t$-test). **f** Immunoblots showing palmitoylation (Streptavidin pull down-palmitoylation also named "proteomic ABE", see Methods) (top) and expression (middle) of zDHHC3 in the hippocampus of SD and HFD mice. Samples without $NH_2OH$ are negative controls. Densitometry (bottom) of palmitoylated zDHHC3/total protein ($n = 6$; statistics by Mann–Whitney test). Data are expressed as mean ± s.e.m. *$p$ < 0.05; **$p$ < 0.01; n.s. not significant. See also Supplementary Fig. 2

To better characterize the insulin resistance-dependent upregulation of PAT, we investigated the hypothesis that FoxO3a transcriptionally regulated zDHHC3. Notably, nutrients modulate FoxO3a interaction with chromatin remodelers and its transcriptional activity[24–26]. Bioinformatic analysis of the mouse zDHHC3 locus (NC_00075.6) revealed the presence of several putative FoxO responsive elements (pFRE) both upstream and downstream of the transcription start site (pFRE1 containing: −2929, −2713, −2696, and −2685; pFRE2 including: −1148; pFR3 containing: +11,067 and +11,363; and pFRE4 including: +12,116). Chromatin immunoprecipitation from hippocampal neuron extracts revealed that FoxO3a bound two of these genomic regions (pFRE1 and pFRE2), and the recruitment on the sequence pFRE2 was significantly increased by IPA treatment (+103 ± 12%; Fig. 3d). Accordingly, IPA increased the transcriptional activation marker lysine 9 histone 3 acetylation on the same regulatory sequences (+75 ± 11% on pFRE1; +129 ± 8% on pFRE2; Fig. 3d).

Next, we asked whether IPA could impinge on GluA1 palmitoylation similarly to HFD. Treatment of hippocampal neurons with IPA (for 1–24 h) induced a time-dependent increase of GluA1 palmitoylation (Fig. 3e) along with inhibition of pGluA1 Ser[845], but no changes in total GluA1 protein amounts ($F_{3.47} = 4.42$, CTR vs. IPA$_{24h}$ $p = 0.018$; Fig. 3f). These results indicated that IPA reproduced in vitro the effects of HFD on both AMPAR GluA1 subunit palmitoylation/phosphorylation and zDHHC3 expression.

**IPA affects GluA1 localization and AMPA currents**. To reveal the biological and functional outcome of IPA-dependent GluA1 hyper-palmitoylation, we investigated the subcellular localization of GluA1 and recorded AMPAR-mediated postsynaptic currents. To determine the surface vs. intracellular protein localization, we harvested hippocampal neurons with or without the addition of

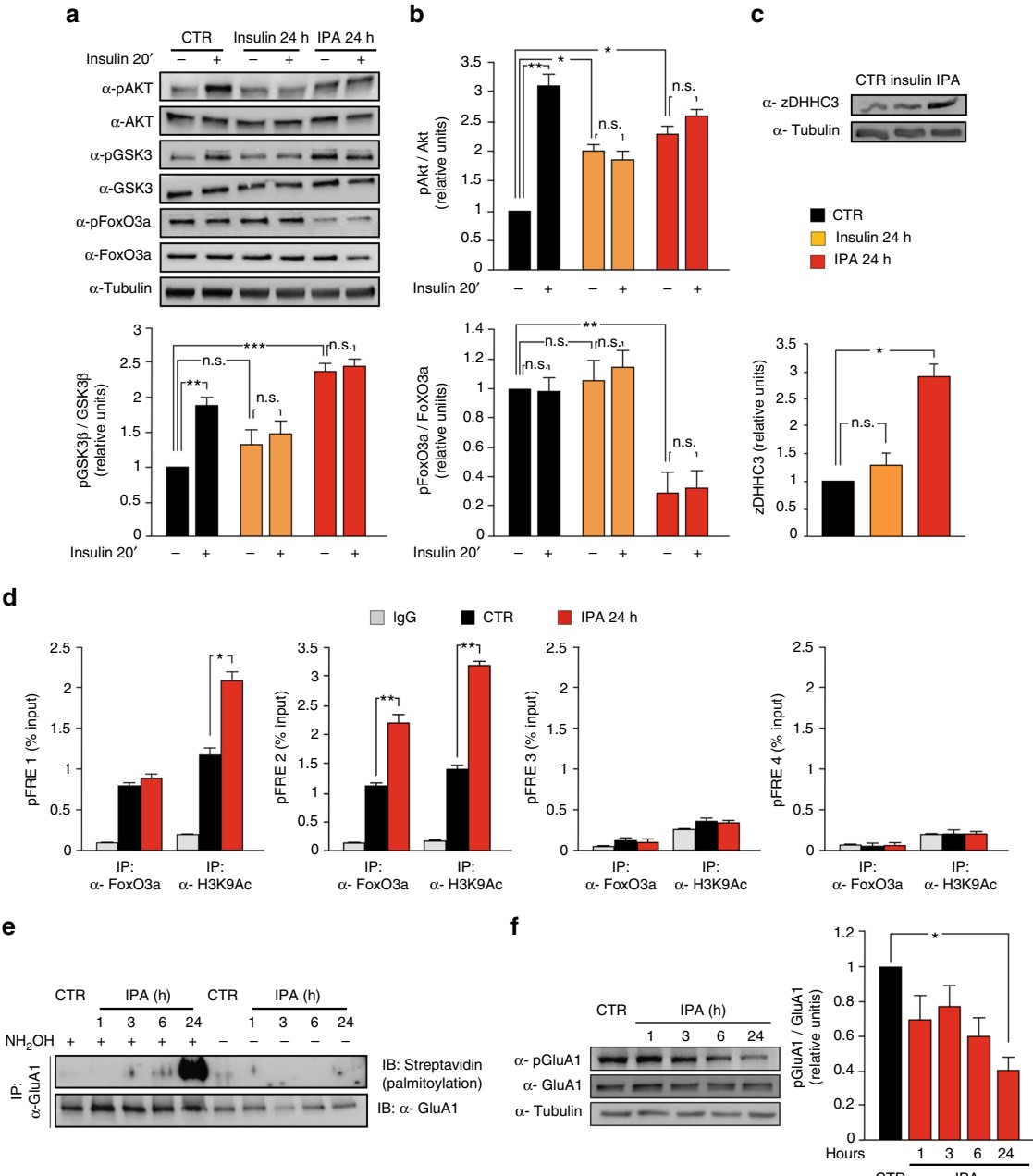

**Fig. 3** Insulin and palmitic acid (IPA) transcriptionally induce zDHHC3 and affect palmitoylation and phosphorylation of GluA1 in hippocampal neurons. **a** Immunoblots of pGSK3β Ser[9] and pFoxO3a Ser[253] after 24 h of insulin or IPA treatment and upon acute stimulation with insulin. **b** Densitometry of pAkt Ser[473] (top), pGSK3β Ser[9] (bottom, left), and pFoxO3a Ser[253] (bottom, right) blots, normalized to both the corresponding total protein and tubulin; experiments were performed in triplicate (statistics by two-way ANOVA and Bonferroni post hoc). **c** Immunoblots (top) and densitometry (bottom) of zDHHC3 expression after insulin or IPA treatment; experiments were performed in triplicate (statistics by one-way ANOVA and Bonferroni post hoc). **d** Chromatin immunoprecipitation assays of FoxO3a binding to and histone H3 lysine 9 acetylation (H3K9Ac) of putative FoxO3a responsive elements (pFRE) around the zDHHC3 promoter in hippocampal neurons treated with vehicle (CTR) or IPA (statistics by Mann–Whitney test). Data represent mean values of three independent experiments. **e** Immunoblots of palmitoylated GluA1 (top) and total immunoprecipitated protein (bottom) in hippocampal neurons. Samples without NH₂OH are negative controls. The experiment was repeated three times with similar results. **f** Immunoblots of pGluA1 Ser[845] (left) and densitometry of pGluA1 Ser[845] normalized to both total GluA1 and tubulin (right). The experiment was repeated three times (n = 3, statistics by one-way ANOVA and Bonferroni post hoc). Data are shown as mean ± SEM. *p < 0.05; **p < 0.01; ***p < 0.001; n.s. not significant

the membrane impermeant cross-linking reagent bis (sulfo-succinimidyl) suberate (BS³). BS³ is a cross-linking agent that forms clusters of GluR subunits expressed on the cell surface[27]. The following SDS-PAGE assay revealed an abundance of GluA1 cytoplasmic monomers (indicated with [C]) and a significant decrease of plasma membrane receptor tetramers containing GluA1 (indicated with [S]) in neurons stimulated with IPA ([C]

+ 48.5 ± 6.7% and [S] −26.8 ± 5.6% vs. control; Fig. 4a, b). We also performed GluA1 immunostaining without membrane permeabilization in order to specifically detect the fraction of receptors localized on the membrane. Accordingly, upon IPA treatment, surface GluA1 content notably decreased in neurites (−44%; Fig. 4c and Supplementary Fig. 3c), whereas GluA1 fluorescence intensity increases in the Golgi apparatus (+25%;

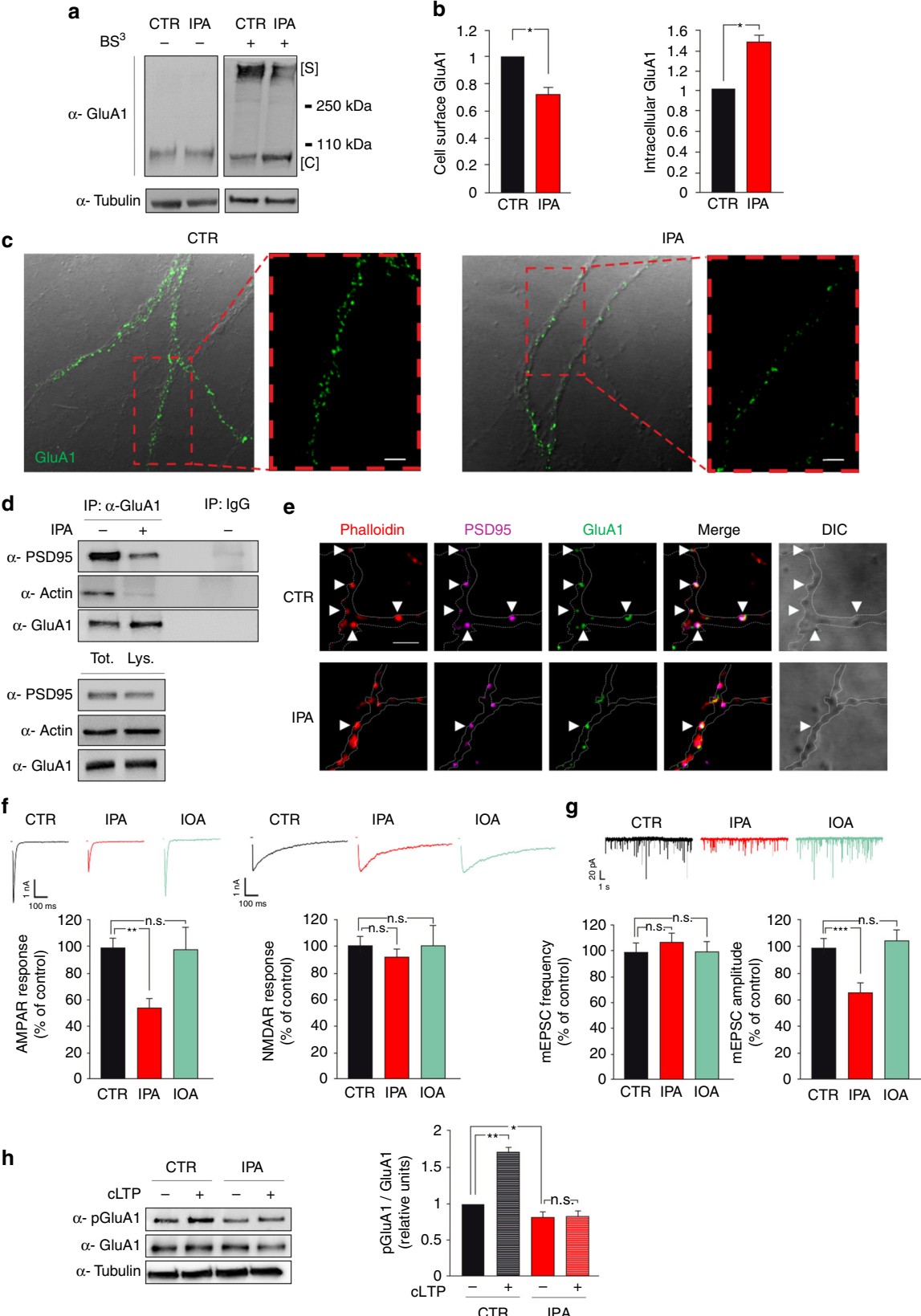

Supplementary Fig. 3d). IPA did not aspecifically inhibit the antibody hybridization as demonstrated by GluA1 immuno-fluorescence in permeabilized neurons (Supplementary Fig. 3a). Next, we investigated whether IPA impaired the binding of GluA1 with synaptic protein complexes. IPA reduced the interaction of GluA1 with the synaptic scaffold protein PSD95 ($-79 \pm 4.1\%$; Fig. 4d and Supplementary Fig. 3b). IPA treatment did not affect the total amount of both GluA1 and PDS95, but rather decreased the stoichiometry of the binding, suggesting either lower affinity between the two proteins or reduced co-

localization in the synaptic membrane. In keeping with the latter possibility, we observed a markedly lower interaction of GluA1 with actin (−74 ± 3.2%; Fig. 4d and Supplementary Fig. 3b).

In addition, we examined GluA1 and PSD95 co-localization by double immunostaining experiments in neurons in which F-actin was labeled with fluorescent phalloidin to visualize dendritic processes. IPA reduced GluA1/PSD95 co-localization (Fig. 4e) without affecting PSD95 total amount at the synapse (Supplementary Fig. 3e) nor PSD95 palmitoylation (Supplementary Fig. 3 f). Collectively, co-immunoprecipitation and immunofluorescence experiments indicated that the IPA-induced GluA1 hyper-palmitoylation inhibited its synaptic membrane localization. Finally, we studied the impact of IPA on glutamatergic synaptic transmission by whole-cell patch-clamp recordings in autaptic hippocampal neurons. After 24-hour treatment with IPA, AMPAR-mediated EPSCs were significantly lower than controls (−44.0 ± 4.3%; $F_{2,59} = 4.971$, $p = 0.009$; Fig. 4f). In keeping with molecular data pointing to specific hyper-palmitoylation of AMPAR subunits, NMDA response was not significantly affected by IPA treatment (Fig. 4f). The spontaneous miniature EPSC (mEPSC) amplitude, but not their frequency, was markedly reduced upon IPA stimulation (−34.8 ± 6.1%; $F_{2,58} = 6.766$, $p < 0.001$; Fig. 4g), consistent with the reduced AMPAR density at the postsynaptic site. To check the specificity of the effects of palmitic acid on synaptic function we treated autaptic hippocampal microcultures with a cocktail of insulin and oleic acid (IOA). After 24-h treatment with IOA, we did not observe any significant changes in evoked AMPAR-mediated and NMDAR-mediated currents (Fig. 4f), mEPSCs (Fig. 4g), nor did we detect any modification of GluA1 palmitoylation with ABE assay (Supplementary Fig. 3g). These findings focused our attention on the critical role of IPA-dependent AMPAR post-translational changes (i.e., increased GluA1 palmitoylation) in the insulin resistance-related synaptic dysfunction.

Recruitment of AMPARs at the postsynaptic site is a pivotal determinant of LTP at excitatory synapses[28].To investigate the effects of IPA on the activity-dependent phosphorylation of GluA1, we used a chemical LTP protocol (cLTP) that reportedly enhances surface expression of GluA1-containing AMPARs in neurons[29]. After confirming the increased pGluA1 Ser[845] upon cLTP in our experimental model (Supplementary Fig. 3h), we tested the effects of IPA on AMPAR subunit activation. IPA treatment markedly reduced basal GluA1 Ser[845] phosphorylation and, most importantly, abolished its cLTP-dependent enhancement ($F_{3,86} = 65.62$, $CTR_{NT}$ vs. $CTR_{cLTP}$ $p = 0.007$; $CTR_{NT}$ vs. $IPA_{NT}$ $p = 0.014$; Fig. 4h), thus suggesting that IPA-dependent unbalance of GluA1 palmitoylation/phosphorylation ratio affects the activity-dependent changes underlying synaptic plasticity.

**zDHHC3 silencing prevents HFD-induced cognitive deficits.** To demonstrate the causative role of zDHHC3/GluA1 pathway in the synaptic plasticity deficit induced by insulin resistance, we performed LTP experiments in hippocampal organotypic slices biolistically transfected with plasmid-encoding shRNA for zDHHC3. LTP at CA3-CA1 synapses was virtually abolished by 24-h IPA treatment in control slices (+14.6 ± 18.4%, $shCTR_{IPA}$ vs. 160.5 ± 23.8%, $shCTR_{VEH}$; Fig. 5a). Silencing of palmitoyl-transferase did not per se affect the LTP magnitude at CA3-CA1 synapses, but it abolished the IPA-dependent LTP impairment observed in controls (+161.9 ± 35.4%, $shzDHHC3_{IPA}$ vs. 156.8 ± 42.9%, $shzDHHC3_{VEH}$; $F_{3,15} = 6.234$, $shCTR_{VEH}$ vs. $shCTR_{IPA}$ $p = 0.0005$; Fig. 5a). Moreover, to deeply investigate the critical role of zDHHC3-mediated hyper-palmitoylation in HFD-dependent cognitive impairment, we specifically inhibited the expression of palmitoyltransferase in mice fed with SD or HFD by intra-hippocampal injection of lentivirus harboring shRNA against zDHHC3 (LV-shzDHHC3) (Supplementary Fig. 4a). Food consumption and effect of diet on body weight, both monitored weekly, were comparable between controls (injected with control shRNA, LV-shCTR) and LV-shzDHHC3 mice (Supplementary Fig. 4b), indicating that neither feeding behavior nor gross energy metabolism were affected in mutant mice. We next checked the levels of zDHHC3 in the hippocampus of animals. HFD induced the expression of zDHHC3 in LV-shCTR mice similarly to non-injected mice (+38.06%, Supplementary Fig. 4c). Conversely, LV-shzDHHC3 mice showed reduced levels of palmitoyltransferase in the hippocampus after both dietary regimens (Supplementary Fig. 4c). The silencing was specifically localized at the hippocampus as indicated by non-detectable difference of zDHHC3 expression in the neocortex of LV-shzDHHC3 mice (Supplementary Fig. 4d). Strikingly, LV-shzDHHC3 mice were resistant to the HFD-dependent cognitive impairment. In particular, in the NOR test, preference for the novel object was clearly impaired by HFD in LV-shCTR mice but not in the LV-shzDHHC3 mice (57.5% vs. 66.9% in LV-shCTR mice, 65% vs. 65.3% in LV-shzDHHC3 mice; $F_{3,008} = 6.35$, LV-shCTR$_{SD}$ vs. LV-shCTR$_{HFD}$ $p = 0.0055$; LV-shCTR$_{SD}$ vs. LV-shzDHHC3$_{HFD}$ $p = 0.34$; Fig. 5b), confirming the pivotal role of zDHHC3 in the detrimental effect of nutrient overload on this cognitive task. Moreover, LV-shzDHHC3 animals fed HFD showed learning curves similar to both LV-shzDHHC3 mice and controls fed SD during the training phase of MWM and spent less time to reach the platform than LV-shCTR mice fed HFD starting from the second day of the training ($F_{3,008} = 9.81$ for day 2, $F_{3,008} = 5.74$ for day 3, and $F_{3,008} = 7.02$ for day 4; LV-shCTR$_{SD}$ vs. LV-shzDHHC3$_{HFD}$ $p > 0.05$ in all days; Fig. 5c). Accordingly, LV-shzDHHC3 mice fed HFD remembered the platform location during the probe test of MWM and spent significantly more time in the target quadrant

**Fig. 4** IPA affects synaptic localization of GluA1 and AMPA currents in hippocampal neurons. **a** Immunoblots of control (−) and BS[3] cross-linked (+) surface exposed receptors upon treatment with vehicle (CTR) or IPA showing cytoplasmic GluA1 monomers (C) and surface subunit tetramers including GluA1 (S). **b** Densitometry of both cell surface (left) and intracellular (right) GluA1 fractions normalized to tubulin; the experiment was repeated six times (statistics by Mann–Whitney test). **c** Immunofluorescence analysis of surface GluA1 in hippocampal neurons. A magnification is shown in the box (right); scale bar = 5 μm. **d** Immunoblots of GluA1 interaction with both PSD95 (top) and actin (middle). On bottom, cell lysates probed with α-PSD95, α-actin, and α-GluA1. The experiment was repeated four times. **e** Confocal images of immunofluorescence double staining of neurites upon IPA treatment. PSD95 (fuchsia) and GluA1 (green) immunoreactivity are merged. Neurites are visualized by phalloidin staining and differential interference contrast image (DIC). Arrows show dendritic spines exhibiting co-localization of GluA1 and PSD95; scale bar = 10 μm. **f** Representative traces (top) and bar graphs showing mean AMPAR (bottom, left) and NMDAR currents (bottom, right) in autaptic neurons exposed to vehicle (CTR), IPA, or insulin and oleic acid (IOA); recordings for AMPAR currents: $n = 21$ per each group (statistics by one-way ANOVA and Student–Newman–Keuls post hoc). **g** Representative traces (top) and bar graphs showing mean mEPSC frequency (bottom, left) and amplitude (bottom, right) in autaptic neurons; mEPSC recordings: $n = 21$ controls, $n = 20$ IPA, $n = 20$ IOA (statistics by one-way ANOVA and Student–Newman–Keuls post hoc). **h** Immunoblots and densitometry of chemical LTP-dependent pGluA1 Ser[845] in hippocampal neurons. Experiment was repeated four times (statistics by two-way ANOVA and Bonferroni post hoc). Data are shown as mean ± SEM *$p < 0.05$; **$p < 0.01$; ***$p < 0.001$; n.s. not significant. See also Supplementary Fig. 3

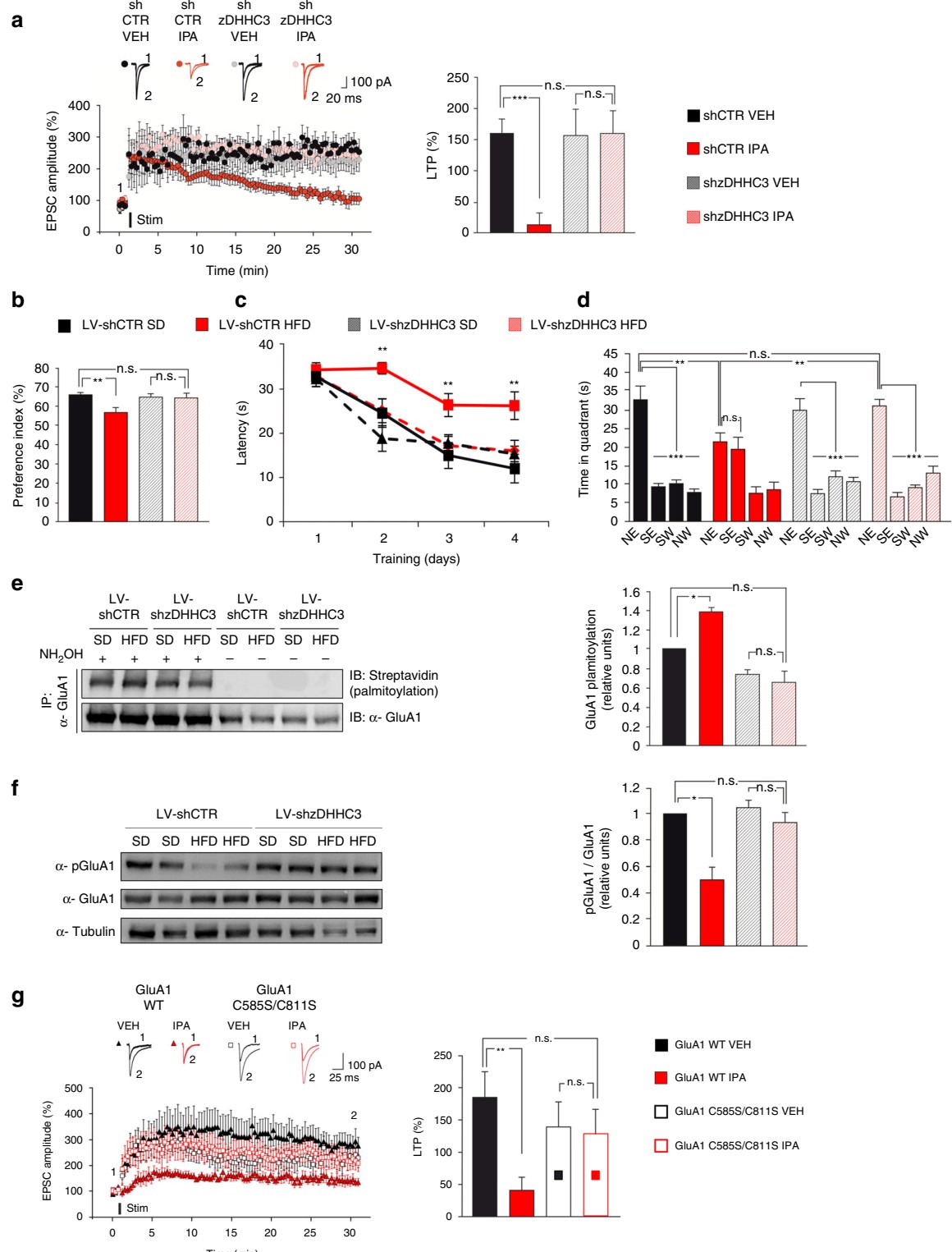

than LV-shCTR mice fed HFD (time in the target quadrant: $F_{3.008} = 4.87$, LV-shCTR$_{SD}$ vs. LV-shCTR$_{HFD}$ $p = 0.004$; LV-shCTR$_{SD}$ vs. LV-shzDHHC3$_{HFD}$ $p = 0.61$; LV-shzDHHC3$_{HFD}$ vs. LV-shCTR$_{HFD}$ $p = 0.003$; time spent in the 4 quadrants: $F_{3.008} = 7.59$ for LV-shCTR$_{HFD}$, NE vs. SE $p = 0.83$; Fig. 5d). Notably, zDHHC3 silencing also abolished both the hyper-palmitoylation (+ 39.7%; $F_{4.75} = 13.64$, LV-shCTR$_{SD}$ vs. LV-shCTR$_{HFD}$ $p = 0.013$; LV-shCTR$_{SD}$ vs. LV-shzDHHC3$_{HFD}$ $p = 0.11$; Fig. 5e) and the

hypo-phosphorylation (−49.6%; $F_{3.49} = 8.55$, LV-shCTR$_{SD}$ vs. LV-shCTR$_{HFD}$ $p = 0.016$; Fig. 5f) of AMPA receptor subunit in the hippocampi of HFD mice. Moreover, biolistic overexpression of palmitoylation-deficient GluA1 (GluA1 C585S/C811S), but not GluA1 wild type (WT), abolished the impairment of LTP induced by IPA in organotypic slices ($F_{2.89} = 3.128$, GluA1 WT$_{VEH}$ vs. GluA1 WT$_{IPA}$ $p = 0.004$; Fig. 5g). Collectively, our findings point to the key role of both zDHHC3 overexpression and AMPAR

hyper-palmitoylation in the cognitive impairment induced by HFD.

## 2-BP rescues synaptic plasticity in mice fed HFD.

To get information potentially useful for the development of pharmacological strategies against cognitive decline in metabolic disease, we tested the effects of the non-specific inhibitor of S-palmitoylation, 2-BP, on synaptic plasticity and cognitive impairment in our models of brain insulin resistance. In hippocampal organotypic slices concurrently treated with IPA and 2-BP LTP was not significantly different from controls (Supplementary Fig. 5a). Notably, 24-h treatment with 5 μM 2-BP alone did not per se affect the LTP magnitude at CA3-CA1 synapses (Supplementary Fig. 5a), nor modified mEPSC frequency, mEPSC amplitude, and AMPA-mediated current density (mEPSC frequency: $0.71 \pm 0.17$ [$n = 14$] and $0.71 \pm 0.07$ Hz [$n = 17$], in vehicle-treated and 2-BP-treated slices, respectively, $p = 0.99$; mEPSC amplitude: $19.1 \pm 2.4$ and $15.6 \pm 1.8$ pA, respectively, $p = 0.26$; AMPA current density: $8.8 \pm 0.4$ and $10.1 \pm 0.3$ pA pF$^{-1}$, $p = 0.48$; statistics by unpaired Student's $t$-test.). Strikingly, ABE assay performed on the same slice preparations showed that 2-BP reverted GluA1 palmitoylation (Supplementary Fig. 5b). Control experiments showed no toxicity of 2-BP under our experimental conditions. Specifically, the percentage of apoptotic neurons after treatment with 5 μM 2-BP for 24 h was not significantly different from controls ($16 \pm 1.7\%$ [$n = 1886$ cells] vs. $14 \pm 1.4\%$ [$n = 1829$ cells], respectively; Supplementary Fig. 5c). Thus, 2-BP appeared to be a drug with the potential to counteract the effects of HFD on brain plasticity.

These findings prompted us to investigate the in vivo efficacy of 2-BP. Mice fed for 6 weeks with either SD or HFD were intranasally injected with saline or 2-BP for the entire duration of the diet. At the end of dietary regimen, we investigated the effect of 2-BP on hippocampal synaptic plasticity, learning, memory, and GluA1 palmitoylation. Interestingly, LTP at CA3-CA1 synapses was completely restored in hippocampal brain slices obtained from mice treated with HFD + 2-BP ($95.8 \pm 11.3\%$, HFD$_{2-BP}$ mice vs. $33.5 \pm 6.4\%$, HFD$_{VEH}$ mice; $F_{2.96} = 10.03$, SD$_{VEH}$ vs. HFD$_{VEH}$ $p = 0.00011$ and HFD$_{2-BP}$ vs. HFD$_{VEH}$ $p = 0.00028$; Fig. 6a). Moreover, in the NOR test HFD$_{2-BP}$ mice showed a significantly higher preference toward the novel object than HFD$_{VEH}$ animals ($68.3 \pm 1.7\%$ vs. $59.2 \pm 0.5\%$; $F_{3.008} = 34.17$, SD$_{VEH}$ vs. HFD$_{VEH}$ $p = 9.16 \times 10^{-7}$ and HFD$_{2-BP}$ vs. HFD$_{VEH}$ $p = 8.67 \times 10^{-6}$; Fig. 6b) that was not significantly different from SD$_{VEH}$ mice. Additionally, co-administration of 2-BP almost completely abolished the detrimental effects of HFD during the training of MWM (day 2: $F_{3.008} = 3.6$, SD$_{VEH}$ vs. HFD$_{VEH}$ $p = 0.0013$, HFD$_{VEH}$ vs. HFD$_{2-BP}$ $p = 0.012$; day 3: $F_{3.008} = 4.06$, SD$_{VEH}$ vs. HFD$_{VEH}$ $p = 0.007$, SD$_{VEH}$ vs. HFD$_{2-BP}$ $p = 0.44$; and day 4: $F_{3.008} = 15.26$, SD$_{VEH}$ vs. HFD$_{VEH}$ $p = 1.7 \times 10^{-3}$, SD$_{VEH}$ vs. SD$_{2-BP}$ $p = 0.907$; HFD$_{VEH}$ vs. HFD$_{2-BP}$ $p = 0.00085$; Fig. 6c). Accordingly, HFD$_{2-BP}$ mice discriminated the target quadrant similarly to controls, and they spent there significantly more time than HFD$_{VEH}$ animals (time in the target quadrant: $F_{3.008} = 5.22$, SD$_{VEH}$ vs. HFD$_{VEH}$ $p = 0.0009$, SD$_{VEH}$ vs. SD$_{2-BP}$ $p = 0.175$; SD$_{VEH}$ vs. HFD$_{2-BP}$ $p = 0.145$; HFD$_{VEH}$ vs. HFD$_{2-BP}$ $p = 0.011$; time spent in the four quadrants: $F_{3.008} = 29.02$ for SD$_{VEH}$, $F_{3.008} = 15.94$ for SD$_{2-BP}$; $F_{3.008} = 10.96$ for HFD$_{VEH}$ NE vs. SE $p = 0.52$, $F_{3.008} = 19.75$ for HFD$_{2-BP}$; Fig. 6d). These findings indicate that, in our experimental conditions, 2-BP rescued both hippocampal synaptic plasticity and hippocampus-dependent memory impairment induced by HFD without affecting LTP, learning, and memory in SD mice. Finally, GluA1 palmitoylation in hippocampal slices obtained from HFD$_{2-BP}$ mice was significantly lower than in HFD$_{VEH}$ slices and not significantly different from controls ($F_{3.86} = 13.27$, SD$_{VEH}$ vs. HFD$_{VEH}$ $p = 0.03$, SD$_{VEH}$ vs. SD$_{2-BP}$ $p = 0.2$, HFD$_{VEH}$ vs. HFD$_{2-BP}$ $p = 0.025$; Fig. 6e). Remarkably, these results were independent of peripheral effects of 2-BP on metabolism, as indicated by comparable insulin levels and weight between HFD$_{VEH}$ and HFD$_{2-BP}$ mice (Supplementary Fig. 5d).

## Discussion

HFD in C57BL/6 mice is a well-established experimental model of obesity and insulin resistance, almost completely resembling the hallmarks of metabolic syndrome identified in humans[30]. It also impacts on brain function and affects synaptic plasticity, learning, and memory through molecular mechanisms that are still poorly understood[31]. Palmitoylation dynamically regulates neuronal protein localization and synaptic function[32]. Essentially, palmitoylation consists of the covalent binding of a palmitic acid molecule to a protein, but is unknown whether diet-dependent or metabolic disease-dependent fatty acid excess can have an impact on protein palmitoylation and alter synaptic function, learning, and memory. Here we show that HFD increases palmitic acid accumulation in the hippocampus of mice (Fig. 1b), induces hippocampal insulin resistance (Fig. 1d, e), and impairs synaptic plasticity (Fig. 1a). Insulin has been reported to interfere with protein palmitoylation in endothelial cells[33] and to underlie the cross talk between metabolic dysfunction and synaptic plasticity impairment[34]. Therefore, our hypothesis was that HFD altered the palmitoylation of neuronal proteins critically involved in synaptic plasticity.

We started testing the palmitoylation of AMPAR and NMDAR subunits because this post-translational modification is known to regulate their trafficking and insertion to neuronal membrane during LTP[32]. We found that HFD specifically impinges on

**Fig. 5** Hippocampal silencing of zDHHC3 abolishes HFD-dependent learning and memory impairment. **a** Time course (left) of LTP at CA3-CA1 synapses in hippocampal organotypic slices transfected with plasmid-encoding for control shRNA or zDHHC3 shRNA and treated with vehicle (VEH) or IPA for 24 h. Results are expressed as percentages of baseline EPSC amplitude (=100%). Insets (top) show representative EPSC at baseline (1) and during the last 5 min of LTP recording (2). On right, mean LTP values during the last 5 min ($n = 7$ for each group; statistics by two-way ANOVA and Bonferroni post hoc). **b** Preference for the novel object of mice fed SD or HFD and injected with lentiviral particles harboring control shRNA (LV-shCTR) or shRNA against zDHHC3 (LV-shzDHHC3) ($n = 9$ for each group; statistics by two-way ANOVA and Bonferroni post hoc). **c** Latency to reach the platform ($n = 9$ for each group; significance is indicated for LV-shCTR$_{HFD}$ vs. all other groups; statistics by two-way ANOVA and Bonferroni post hoc). **d** Time spent in the four quadrants during probe test. NE is the target quadrant ($n = 9$ for each group; statistics by two-way ANOVA and Bonferroni post hoc). **e** Palmitoylated GluA1 (left, top) and total immunoprecipitated protein (left, bottom) in hippocampi. Densitometry (right) of palmitoylated GluA1/total immunoprecipitated GluA1 ratio ($n = 3$ per each group; statistics by two-way ANOVA and Bonferroni post hoc). **f** Immunoblots of pGluA1 Ser$^{845}$ and densitometry of pGluA1 Ser$^{845}$ normalized to both the total GluA1 and tubulin ($n = 5$ mice per group; statistics by two-way ANOVA and Bonferroni post hoc). **g** Time course (left) of LTP at CA3-CA1 synapses in hippocampal organotypic slices transfected with plasmids encoding for GluA1 WT or GluA1 C585S/C811S. Results are expressed as percentages of baseline EPSC amplitude (=100%). Insets (top) show representative EPSC at baseline (1) and during the last 5 min of LTP recording (2). On right, mean LTP values during the last 5 min ($n = 12$ for each group; statistics by two-way ANOVA and Bonferroni post hoc). Data are expressed as mean ± SEM *$p < 0.05$; **$p < 0.01$; ***$p < 0.001$; n.s. not significant. See also Supplementary Fig. 4

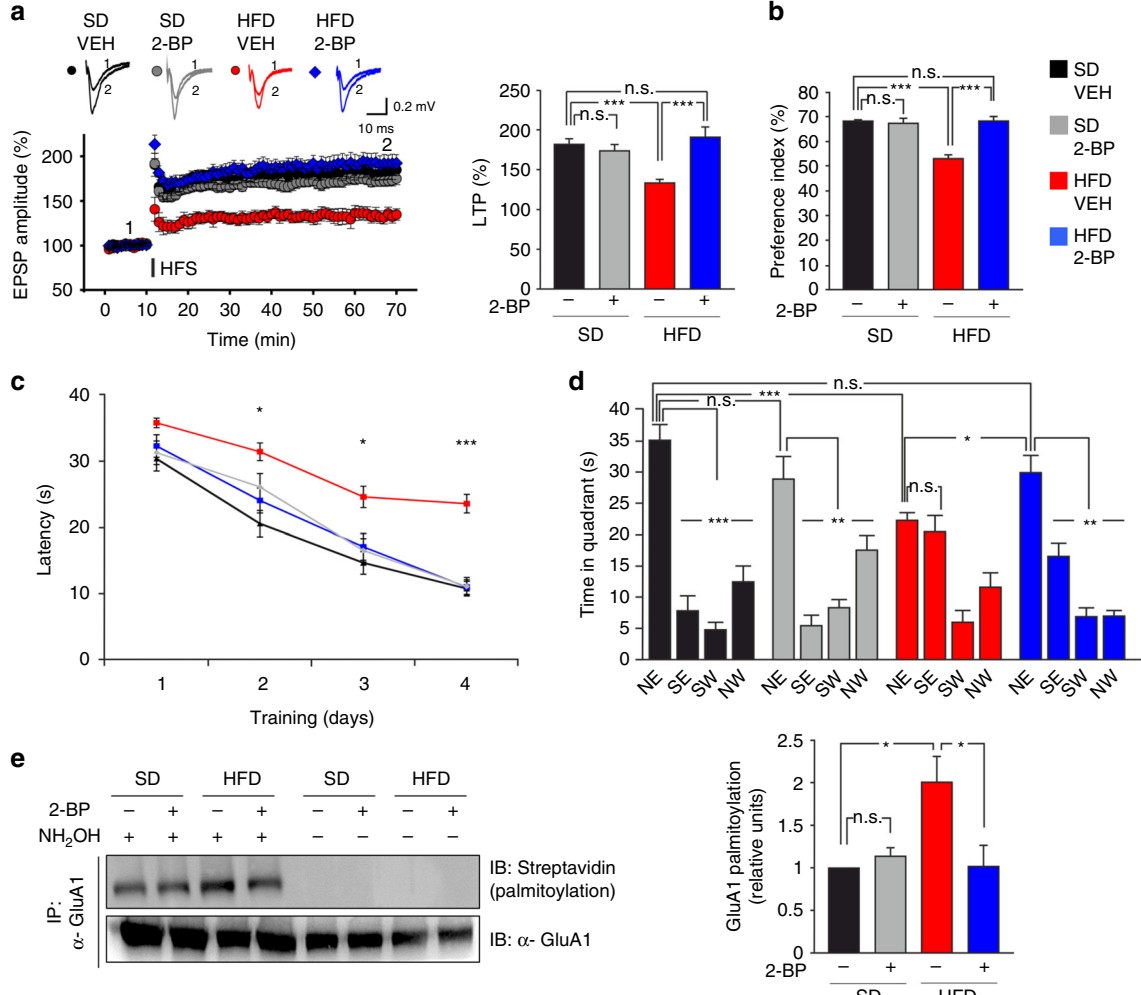

**Fig. 6** 2-BP reverts GluA1 palmitoylation and rescues both synaptic plasticity impairment and memory loss induced by HFD. **a** Time course (left) of LTP at CA3-CA1 synapses induced by HFS delivered at time 10 (line) in hippocampal slices of mice fed with SD or HFD for 6 weeks and intranasally injected with vehicle or 2-BP (SD$_{VEH}$, SD$_{2-BP}$, HFD$_{VEH}$, HFD$_{2-BP}$; n = 12 slices per each group). Results are expressed as percentages of baseline EPSP amplitude (=100%). Insets (top) show representative EPSPs at baseline (1) and during the last 5 min of LTP recording (2). On right, LTP recorded during the last 5 min (statistics by two-way ANOVA and Bonferroni post hoc). **b** Preference for the novel object in NOR paradigm (n = 9 for each group; statistics by two-way ANOVA and Bonferroni post hoc). **c** Latency to reach the hidden platform in MWM test (n = 9 for each group; significance is indicated between SD$_{VEH}$ or HFD$_{2-BP}$ and HFD$_{VEH}$ mice; statistics by two-way ANOVA and Bonferroni post hoc). **d** Time spent in the four quadrants during probe test of MWM test. NE is the target quadrant (n = 9 for each group; statistics by two-way ANOVA and Bonferroni post hoc). **e** Immunoblots (left) of palmitoylated GluA1 (top) and total immunoprecipitated protein (bottom) in hippocampi of SD and HFD mice. On right, densitometry of palmitoylated GluA1/total immunoprecipitated GluA1 amount ratio (n = 4; statistics by two-way ANOVA and Bonferroni post hoc). Data are expressed as mean ± SEM *p < 0.05; **p < 0.01; ***p < 0.001; n.s. not significant. See also Supplementary Fig. 5

palmitoylation of GluA1 and GluA2, but not other glutamate receptor subunits (Fig. 2a, b and Supplementary Fig. 2a,b), thus suggesting that HFD-dependent hyper-palmitoylation depends on specific enzyme activation. HFD transcriptionally induced the acyltransferase zDHHC3 (which targets GluA1) (Fig. 2e) and promoted its palmitoylation (Fig. 2f) in the hippocampus. To identify the metabolic signals affecting AMPAR palmitoylation and deeply investigate their functional outcomes, we set up an in vitro model of insulin resistance. We found that exposure of hippocampal neurons to IPA reproduced both the insulin signaling alterations (Fig. 3a, b) and the overexpression of zDHHC3 (Fig. 3c) that we observed in the hippocampus of HFD mice. The critical step seems to be the dephosphorylation/activation of FoxO3a that regulates the nuclear localization of this transcription factor[35]. FoxO activity is closely related to the insulin and fatty acid response in tissues[36, 37]. We demonstrated that, upon IPA stimulation, FoxO3a was hypo-phosphorylated (Fig. 3a) and

it bound more avidly a regulatory sequence on the zDHHC3 promoter (Fig. 3d). The transcriptional regulation of zDHHC3 by FoxO3a and its post-translational activation add novel elements to the complex modulation of synaptic function by insulin[38]. Insulin may directly stimulate AMPA receptor phosphorylation[39] and endocytosis[40]. Additionally, phosphotidylinositide-3-kinase, which is the main arm of insulin signaling, increases cell surface expression of AMPARs along with LTP[41]. However, we demonstrated that application of IPA, but not insulin alone or IOA, increased both the expression/activation of zDHHC3 (Fig. 3c) and the palmitoylation of GluA1 (Supplementary Fig. 3g). Collectively, this first set of data indicates that both the increase of substrate availability (i.e., the palmitic acid) and the increased activity of zDHHC3 are required for the HFD-related hyper-palmitoylation of GluA1. We also found that the IPA-dependent GluA1 hyper-palmitoylation decreased its phosphorylation (Fig. 3f) and its surface membrane localization (Fig. 4a–c),

thereby inhibiting AMPAR response at the postsynaptic level (Fig. 4f, g) under both basal conditions and following activity-dependent stimulation.

GluA1 delivery to dendritic spines is crucial for LTP induction[42, 43], as also demonstrated by GluA1 knockout mice lacking LTP[44] and showing spatial memory deficits[45]. Accordingly, we showed that LTP at CA3-CA1 synapses was impaired in both organotypic hippocampal slices treated with IPA and brain slices obtained from hippocampi of HFD mice (Supplementary Fig. 5a and Fig. 6a, respectively). The genetic blockade of zDHHC3 in both organotypic brain slices treated with IPA and hippocampi of HFD mice abolished the LTP (Fig. 5a) and memory deficits (Fig. 5b–d), respectively. More importantly, the overexpression of GluA1 mutant lacking the palmitoylation sites in organotypic slices almost completely annulled the detrimental effects of insulin resistance on synaptic plasticity (Fig. 5g). In future studies we plan to validate our findings in a palmitoylation-deficient GluA1 mouse model. Finally, treatment of HFD mice with the non-specific palmitoylation inhibitor, 2-BP, completely rescued hippocampal synaptic plasticity and memory deficits (Fig. 6a–e), thus suggesting the potential efficacy of 2-BP and/or other more specific drugs targeting zDHHC3 in metabolic-dependent cognitive impairment, as well as in other neurological diseases associated with altered plasticity involving AMPARs. Certainly, 2-BP may act on numerous targets and modulate the palmitoylation of several synaptic proteins, leading to changes that may either promote (in case they occur on GluA1 or GluA2) or inhibit synaptic plasticity (in case they occur on PSD95 or GABAA receptor γ2). The scenario is also more complex because palmitoylation of different cysteine residues in the same target may induce opposite effects (e.g., NR2A or NR2B)[9], and there is evidence that 2-BP may also inhibit depalmitoylating enzymes such as thioesterases[46]. Consequently, it is very difficult to predict the net result of many, and potentially conflicting, effects exerted by 2-BP on different targets involved in the establishment of LTP. We cannot rule out that the effects of HFD on hippocampal synaptic plasticity may partly depend on altered palmitoylation of other zDHHC3 targets, as suggested by the increased palmitoylation of GabaA Rγ2 (Supplementary Fig. 2b,c). Besides affecting the palmitoylation of several neuronal proteins, 2-BP also induces other effects including modulation of fatty acid β-oxidation and NADPH cytochrome c reductase activity[47],[48]. Collectively, our findings suggest that both HFD and IPA hinder hippocampal plasticity and hippocampal-dependent learning and memory by reducing AMPAR trafficking at synapses because of zDHHC3-dependent hyper-palmitoylation of GluA1. Our study adds a new layer to AMPAR and synaptic plasticity regulation by nutrient-related signals and propose a novel molecular circuitry triggered by brain insulin resistance and involving epigenetic/post-translational regulation of zDHHC3, potentially linking metabolic and neurodegenerative diseases. Emerging evidence suggests that zDHHCs are associated with several neurological disorders[32]. Future studies are necessary to determine the contribution of metabolic regulation of protein palmitoylation to the age-related and diet-related cognitive decline.

## Methods

**Animals.** Male C57BL/6 mice (30–35 days old), derived from Animal Facility of Catholic University, were used and randomly assigned to two feeding regimens: (i) standard diet (SD, control) and (ii) HFD. Different groups of mice were used for each experimental test. Mice were always housed in group (3–5 animals per cage), except after stereotaxic injection when they were singularly housed. All animal procedures were approved by the Ethics Committee of the Catholic University and were fully compliant with Italian (Ministry of Health guidelines, Legislative Decree No. 116/1992) and European Union (Directive No. 86/609/EEC) legislations on animal research. The methods were carried out in strict accordance with the approved guidelines. The animals were housed under a 12-h light-dark cycle at

room temperature (RT: 19–22 °C), fed with their respective diet and water ad libitum and body weight was weekly monitored.

**Diet and drug administration.** Mice from the same litter were randomly assigned to different experimental groups. Animals were fed with SD or HFD (whose caloric intake was composed by 60% of saturated fatty acids) for 6 weeks. The diets were from Mucedola (Italy). For drug administration experiments, mice were intranasally injected with saline or 2-BP (0.125 nMol per nostril, three times per week) for the entire duration of the diet. For western blotting experiments the mice were starved for 14–16 h before intranasal stimulation with insulin.

**Ex vivo electrophysiology on hippocampal slices.** All experiments were performed on 10–11-week-old male C57BL/6 mice as previously described[49]. Local field potentials (LFPs) were elicited in the CA1 area by placing a bipolar concentric stimulating electrode (FHCNeural microTargeting Worldwide) in the Schaffer collateral pathway. The electrode was connected to a current stimulus isolator (World Precision Instruments). A low impedance glass pipette (1–2 MΩ) was filled with ACSF and placed immediately below the CA1 stratum pyramidale. Recordings were performed in current clamp $I = 0$ mode, using a Multiclamp 700B/Digidata 1550 A system (Molecular Devices). First, the input–output relationship was constructed and the stimulus intensity resulting in 30% of maximal response amplitude was found. After achieving a stable baseline response, LTP was induced by using the high-frequency stimulation protocol (one train of stimuli at 100 Hz, lasting 1 s, repeated four times with an inter-train interval of 10 s). After LTP induction, LFP response amplitude was monitored for at least 60 min. Data were analyzed as previously described[49].

**Whole-cell patch-clamp recordings on autaptic cultures.** Autaptic hippocampal neurons were prepared as previously described[50] and studied from 14 to 21 DIV. Recordings were considered stable when the series and input resistances, resting membrane potential, and stimulus artifact duration did not change >20%. Recordings were obtained with an Axopatch 200B amplifier (Molecular Devices), and stimulation and data acquisition were performed with a Digidata 1200 series digital interface and Clampex 10.2 software (Molecular Devices). EPSCs were recorded in whole-cell mode during continuous perfusion with Tyrode's solution containing 4 mM $Ca^{2+}$, while voltage clamping neurons at −70 mV, with stimuli mimicking action potentials (2 ms at 0 mV) delivered every 20 s. NMDA currents were evoked during continuous perfusion with 4 mM $Ca^{2+}$, $Mg^{2+}$-free Tyrode's solution containing 10 µM of the AMPA receptor blocker NBQX (Tocris Bioscience). mEPSCs were recorded at −70 mV in 60-s epochs. All experiments were performed at RT. Data were analyzed as previously described[50].

**Hippocampal slice cultures.** Hippocampal organotypic slice (350 µm) cultures were prepared from postnatal day 4–7 rats through a McIllwain tissue chopper as described by Kim and colleagues[51]. Plasmids were biolistically transfected into slices at DIV 4–5 by using Gene-Gun (Bio-Rad, CA, USA). GluA1 plasmids were transfected together with a plasmid encoding enhanced green fluorescent protein (EGFP) to identify the transfected neurons. LTP experiments were performed 2–4 days later.

**ELISA assay.** Plasma insulin concentration was determined by using a commercially available Elisa kit (Immunological Sciences). Blood samples were collected from the retro-orbital plexus with sterile glass Pasteur pipettes. After centrifugation, plasma was separated, and stored at −80 °C until further use. The assay was performed according to the manufacturer's instructions.

**Sample preparation and GC-FID analysis.** Lipid extraction from the brain tissue was carried out according to Bligh and Dyer method[52]. The hippocampus samples (15 mg) were homogenized in a glass potter and treated with 3 mL of $CH_2Cl_2$: $CH_3OH$ mixture (1:2 v/v). Then 1 mL of $CH_2Cl_2$ and 1 mL of distilled water were added and the resulting mixture was stirred for 30 s. The organic layer was filtered using PTFE microfilters (pore size is 0.45 µm), and the solvent distilled off in vacuum to give a residue of about 2 mg for hippocampus. Finally, a weighted amount of octanoic acid C8:0 (internal standard, IS) was added to the extract. Samples suitable for the GC-FID analysis were prepared by converting the fatty acid (FA) into the corresponding methyl esters (FAME) by treatment of sample (2–15 mg) with 17 % $BCl_3$ hexane solution (1 mL) according to the method by Morrison and Smith[53]. The hexane solution was left at 90 °C for 1 h, and then concentrated to a final volume of 0.2 mL before the GC-FID analysis.

The GC-FID analyses were carried out using a Thermo Scientific FOCUS GC equipped with a FID detector, split/splitless injector, and HP-Innowax capillary column (30 m × 0.25 mm I.D., 0.25 µm film thickness) from Agilent Technologies. Oven temperature was programmed with a ramp from 100 to 240 °C at 10 °C min$^{-1}$ and then at 240 °C for 20 min. Injector and detector temperatures were set at 240 and 250 °C, respectively. Helium was employed as carrier gas with a flow rate of 1.7 mL min$^{-1}$; 1 µL aliquots were injected and the split ratio of 1:10 was used. Data acquisition was carried out using Chrom-Card Data System. Identification of the

chromatographic peaks was made by comparing the retention times of commercial standards.

**Reagents and standards**. Fatty acid standards, namely octanoic acid (C8:0), C8:0 methyl ester, palmitic acid (C16:0), palmitoleic acid, stearic acid (C18:0), oleic acid (C18:1n-9), and Supelco 37 Component FAME Mix were purchased from Sigma-Aldrich. All solvents and reagents used for the sample preparation and fatty acid dosage were from Sigma-Aldrich. Boron Trichloride solution (ca. 17% in Hexane, ca. 1.0 mol per L) was from TCI Europe. 2-bromohexadecanoic acid (2-BP) was from Santa Cruz. For chemical LTP induction, the cells were incubated with strychnine 0.01 mM, picrotoxin 0.05 mM, glycine 0.3 mM, and tetrodotoxin 0.001 mM. Insulin was from Sigma-Aldrich. For in vitro experiments we used: 20 nM insulin, 0.2 mM palmitic acid, 0.2 mM oleic acid, and 5 μM 2-BP. Lentiviral particles harboring shRNA against mouse zDHHC3 and shRNA control were from Santa Cruz Biotechnology. Plasmid-encoding for control shRNA, zDHHC3 shRNA, and EGFP were from Origene. Plasmid-encoding for GluA1 WT and GluA1 C585S/C811S were a gift from Prof. Richard L. Huganir (The Johns Hopkins University, Baltimore, MD).

**Western blotting**. Cells and tissues were lysed in ice-cold lysis buffer (NaCl 150 mM, Tris-HCl 50 mM pH 8, and EDTA 2 mM) containing 1% Triton X-100, 0.1% SDS, 1× protease inhibitor cocktail (Sigma-Aldrich), 1 mM sodium orthovanadate (Sigma-Aldrich), and 1 mM sodium fluoride (Sigma-Aldrich). The cells were incubated for 15 min on ice with occasional vortexing, spun down at 22,000 × g, 4 °C, and supernatant quantified for protein content (DC protein assay; Bio-Rad). Equal amounts of protein were diluted in Laemmli buffer, boiled, and resolved by SDS-PAGE. The primary antibodies (available in Supplementary Table 2) were incubated overnight and revealed with HRP-conjugated secondary antibodies (Cell Signaling Technology Inc., Danvers, MA). Anti GluA1 (from Millipore) and anti-PSD95 (from Cell Signaling) were diluted 1:1000. Protein expression was evaluated and documented by using UVItec Cambridge Alliance. Images shown were cropped for presentation with no manipulations. The uncropped blot scans of these experiments are shown in Supplementary Figs. 6–10.

**Acyl-biotinyl exchange assay**. ABE was performed as described by Brigidi et al. with minor changes[54]. Briefly, cells and tissues were lysed in lysis Buffer A (150 mM NaCl, 50 mM Tris-HCl, 1% NP-40, and 1% Triton X-100, pH 7.5) with 1× protease inhibitor cocktail (Sigma-Aldrich), 1 mM sodium orthovanadate (Sigma-Aldrich), 1 mM sodium fluoride (Sigma-Aldrich), and 10 mM of N-ethylmaleimide (NEM, Sigma-Aldrich) freshly prepared. Samples were sonicated for 10 s on ice and spun down at 22,000 × g, 4 °C. After the centrifugation, the supernatant was transferred to a new tube, while a small volume of Buffer A containing 1% SDS was added to the pellet. The samples were again sonicated for 10 s on ice and spun down at 22,000 × g, 4 °C. The novel supernatant was added to the one recovered from the first lysis. The lysates were precleared for 30 minutes with empty protein G-sepharose 4B beads (Sigma-Aldrich) before adding 1–2 μg of specific primary antibodies. The samples were incubated overnight at 4 °C with rotating mixer, and then re-incubated with fresh protein G matrix for 3 h at 4 °C. Each sample was split in two halves. One of them was incubated with a buffer containing HAM (Lysis Buffer B: 150 mM NaCl, 50 mM Tris-HCl, 1 M hydroxylamine, 1% NP-40, 1% Triton X-100, and 1 mM biotin-BMCC, pH 7.2). Subsequently, both were incubated for 1 h at 4 °C. The samples were briefly washed and incubated with biotin-BMCC (lysis buffer C: 150 mM NaCl, 50 mM Tris-HCl, 10 μM biotin-BMCC, 1% NP-40, and 1% Triton X-100, pH 6.2) for 2 h at 4 °C in a rotating mixer. The samples were then washed once in Buffer B and twice in Buffer A. Beads were finally resuspended in 30 μL of 1× Laemmli buffer and heated at 90 °C for 5 min. Eluted proteins were resolved using SDS-PAGE and immunoblotting.

A variant of the ABE assay (named "proteomic ABE") was performed as described by Wan et al.[55]. Tissues were lysed in a buffer containing 150 mM NaCl, 50 mM Tris, 5 mM EDTA, pH 7.4, 0.2 % SDS, and 1.7 % Triton X-100, with 1× protease inhibitor cocktail (Sigma-Aldrich) and 10 mM NEM. After 3 chloroform-methanol precipitations, pellets were resuspended in a buffer containing 4% SDS, 0.7 M hydroxylamine and incubated for 1 h at room temperature. After a chloroform-methanol precipitation, the pellets were resuspended in a buffer containing EZ-Link HPDP-Biotin and incubated for 1 h at room temperature. Unreacted HPDP-biotin was removed by chloroform-methanol precipitation and pellets were resuspended in lysis buffer. Samples were diluted to 0.1% SDS and biotinylated proteins were affinity-purified using streptavidin-agarose beads. Beta-mercaptoethanol (1%) was used to cleave HPDP-biotin and release biotinylated proteins from the beads. Proteins were then denatured in sample buffer and analyzed using SDS-PAGE. A list of the antibodies is available in Supplementary Table 2. The uncropped blot scans of these experiments are shown in Supplementary Figs. 7–11.

**Real-time PCR**. Quantitative real-time PCR (qRT-PCR) amplifications were performed using Power SYBRR Master Mix on AB7500 instrument (Life Technologies) according to the manufacturer's instructions. The thermal cycling profile featured a pre-incubation step of 94 °C for 10 min, followed by 40 cycles of denaturation (94 °C, 15 s), annealing (55–57 °C, 30 s), and elongation (72 °C, 20 s).

Melting curves were subsequently generated by heating amplified products at 94 °C for 15 s, cooling to 50 °C for 30 s, followed by slow heating to 94 °C in increments of 0.5 °C.

Melting-curve analyses confirmed that only single products had been amplified. All data were analyzed by comparing to the amplification levels of the actin; ROX was used in the SYBR master mix as reference dye. The threshold values determined by the software were used to calculate transcript expression levels employing the cycle-at-threshold (Ct) method. The data are expressed as fold changes (compared to control) for each amplicon, using the 2-ΔΔCt approach. The primer list is shown in Supplementary Table 1.

**Chromatin immunoprecipitation**. Chromatin immunoprecipitation (ChIP) assays were performed as previously described[56]. Cells lysates were resuspended in 200 μl lysis buffer containing 1% SDS, 50 mM Tris-HCl pH 8.0, and 10 mM EDTA and sonicated on ice with six 10-s pulses with a 20-s interpulse interval. Sample debris was removed by centrifugation and supernatants were precleared with protein G-Sepharose 4B beads (Sigma-Aldrich) for 1 h at 4 °C. About 2 μg of specific antibody or control IgG were added overnight at 4 °C. Immune complexes were collected by incubation with protein G-Sepharose 4B beads for 2 h at 4 °C. After seven washes, immunoprecipitated complexes were separated from beads by vortexing in 150 μL of elution buffer (1% SDS and NaHCO₃ 0.1 M; pH 8.0). An aliquot of 13 μL of NaCl were added to the separated lysate and the samples were decross-linked by incubating them overnight at 65 °C. Chromatin fragments were extracted with PCR DNA fragments purification kit (Geneaid). Primers used for zDHHC3 promoter analysis are shown in Supplementary Table 1.

PCR conditions and cycle numbers were determined empirically, and each PCR reaction was performed in triplicate. Data are expressed as percentage of input calculated by the "Adjusted input value" method according to the manufacturer's instructions (Thermo Fisher scientific ChIP analysis). To calculate the adjusted input the Ct value of input was subtracted by 6.644 (i.e., log2 of 100). Next, the percent input of control and IPA samples was calculated using the formula: $100 \times 2^{(\text{adjusted input} - \text{Ct(ChIP)})}$. The percent input of IgG samples was calculated using the formula $100 \times 2^{(\text{adjusted input} - \text{Ct(IgG)})}$.

**Co-immunoprecipitation**. Cells were lysed in IP buffer (KCl 50 mM, Tris-HCl 50 mM pH 8, EDTA 10 mM, and 1% Nonidet P-40) and part of the lysate was used for total input. The lysates were precleared for 30 min with empty protein G-sepharose 4B beads before being incubated with 1–2 μg of specific antibody or IgG (negative control) and fresh protein G matrix. After 6 h of incubation at 4 °C with rotating mixer, protein G bound immune complexes were collected by centrifugation (22,000 × g, 1 min) and washed six times with IP buffer. Beads were finally resuspended in 1× Laemmli buffer and boiled for 5 min. Eluted proteins were resolved by SDS-PAGE and immunoblotting. The uncropped blot scans of these experiments are shown in Supplementary Fig. 9.

**Primary cultures of hippocampal neurons**. Hippocampal neurons from E17-E19 C57BL/6 mice brains were prepared according to standard procedures[57]. Briefly, hippocampal neurons were incubated for 10 min at 37 °C in trypsin-EDTA solution (0.025%/0.01% w/v; Biochrom AG) in 1× PBS, and tissues were mechanically dissociated at RT with a fire polished Pasteur pipette. Cell suspension was harvested and centrifuged at 600 × g for 5 min. The pellet was resuspended in MEM (Biochrom AG) containing 5% fetal bovine serum, 5% horse serum, 2 mM glutamine, 1% penicillin-streptomycin antibiotic mixture (Sigma-Aldrich), and 25 mM glucose. The cells were plated on poly-L-lysine (0.1 mg mL⁻¹; Sigma-Aldrich) pre-coated wells. After 72 h, medium was replaced again with glutamine-free medium and experiments were performed at 12–14 days in vitro (DIV).

**Vitality assay**. The percentage of apoptotic cells was evaluated by Vybrant® DyeCycle Violet Kit (Thermo Fisher) using a confocal laser scanning system (A1 MP, Nikon, Tokyo, Japan). A 6-h treatment with 200 μM H₂O₂ was used as positive control.

**Immunocytochemistry**. Hippocampal neurons were fixed in PBS solution (4% PFA, 4% sucrose, pH 7.4; Sigma-Aldrich) for 15 min at RT. Immunocytochemistry for surface GluA1 staining was performed as described by Leshchyns'ka et al. with minor changes[58]. After fixation, non-specific staining was blocked in 5% normal goat serum (NGS) at RT for 1 h. Cells were incubated overnight at 4 °C with a primary antibody that recognizes an extracellular epitope of GluA1 (anti GluA1 from Cell Signaling) and then for 90 minutes at RT with the secondary antibody.

To study endogenous GluA1 and PSD95, co-localization of GluA1 and PSD95 and GluA1 accumulated in the Golgi, hippocampal neurons, upon IPA treatment for 24 h, were permeabilized with 0.3% Triton X-100 (Sigma-Aldrich) for 15 min, blocked for 60 min, and then incubated overnight with primary antibodies at 4 °C (anti GluA1 1:500 from Millipore and anti-PSD95 1:500 from Cell Signaling). Cells were then incubated for 90 min at RT with a mixture of secondary antibodies. To reveal filamentous actin, the cells were incubated with rhodamine-conjugated phalloidin for 30 min at RT. Finally, nuclei were counterstained with 4′,6-diamidino-2-phenylindole (DAPI, 0.5 μg per mL for 10 min; Thermo Fisher), and

the cells were coverslipped with ProLong Gold anti-fade reagent (Thermo Fisher). A list of the antibodies is available in Supplementary Table 2.

Images of 512 × 512 pixels were obtained with either an A1 MP, Nikon confocal microscope (Tokyo, Japan) or a Leica Microsystem TCS-SP2 confocal microscope (Wetzlar, Germany) equipped with 40× and 63× magnification oil immersion objectives (numerical aperture 1.4), respectively, plus additional magnification of 3.5 or 5×. Fluorescent dyes were excited at 488, 543, and 633 nm. GluA1 and PSD95 puncta were counted under blinded conditions in randomly chosen segments (length: 15–170 μm) of secondary dendrites from apical branches and expressed as mean number of puncta per 100 μm. A total length of at least 1.5 mm was analyzed for each experimental condition. Puncta were counted using the Image J software (available at http://rsbweb.nih.gov/jj/).

Immunofluorescence for PSD95 was quantified as the sum of fluorescence intensities measured for every pixel in the recorded field. We calculated the "PSD95 density" as the total fluorescence intensity of PSD95 labeling divided by the total area in the field that was occupied by neurons (identified by MAP2 immunoreactivity and calculated by Image J). GluA1 immunoreactivity in the Golgi was quantified as the mean fluorescence intensity evaluated in regions of interest (ROIs) drawn around Golgi apparatus. All experiments were repeated at least three times and at least 10–20 randomly chosen microscopic fields were analyzed for each condition.

**Surface receptor cross-linking assays**. Surface expression of GluA1 was assayed using the membrane-impermeable cross-linking reagent bis(sulfosuccinimidyl) suberate ($BS^3$). Briefly, cells were washed in 1x PBS and incubated for 10 min at 37 °C with 2 mM $BS^3$ (Pierce) in GIBCO® Hank's Balanced Salt Solution (HBSS). The reaction was stopped adding 20 mM glycine (10 min, 4 °C). The cells were then washed twice in 1× PBS and lysed to obtain protein homogenate. Then, they were analyzed directly by SDS-PAGE (4–15% Mini-PROTEAN® TGX™ Precast Gels, Bio-Rad). The uncropped blot scans of these experiments are shown in Supplementary Fig. 9.

**Behavioral experiments**. All behavioral tests were performed from 9 a.m.to 4 p. m., and the data analyzed in blind using an automated video tracking system (Any-Maze™)[56]. Experimental groups were: (i) SD and HFD in Fig.1 and Supplementary Fig. 1; (ii) LV-shCTR SD, LV-shCTR HFD, LV-shzDHHC3 SD, and LV-shzDHHC3 HFD in Fig. 5; and (iii) SD VEH, SD 2-BP, HFD VEH, and HFD 2-BP in Fig. 6. Recognition memory was assessed by novel object recognition test. On day one, animals were familiarized for 10 min to the test arena (45 × 45 cm). On day 2 (training session), they were allowed to explore two identical object placed symmetrically in the arena for 10 min. On day 3 (test session), a new object replaced one of the old objects. The animals were allowed to explore for 10 min and a preference index, calculated as the ratio between time spent exploring the novel object and time spent exploring both objects, was used to measure recognition memory.

Spatial learning and memory were assessed using the Morris water maze test. A circular plastic pool (127 cm in diameter) filled with water colored with nontoxic white paint to obscure the location of an hidden platform was used as experimental apparatus. The pool was ideally divided into four equal quadrants (NE, corresponding to the target quadrant, SE, NW, and SW) and the platform (10 × 10 cm) was placed at the center of the target quadrant in a fixed position. Visual cues were placed on the walls around the pool to orient the mice. The animals were trained for four days, six times a day, and a probe test was administered 24 h after the last training day. Starting positions were varied daily and latencies to reach the platform were recorded. In the probe test, the platform was removed and time spent in each quadrant was measured (60 s of test duration).

**Stereotaxic injection**. Mice were anaesthetized with intraperitoneal injection of a cocktail of ketamine and xylazine and placed in a stereotaxic apparatus (Stoelting Co, Wood Dale, IL). The head skin was cut longitudinally and pedestals with fixed double guide cannulae (C235G-3.0, Plastics One,Inc) directed at the dorsal hippocampus were attached to the calvarium with carboxylate cement (3 M ESPE, Durelon, 3 M Deutschland GmbH, Germany). The following coordinates with lambda and bregma in the same horizontal plane were used: posterior to bregma 2.46 mm; lateral to midline ± 1.5 mm. Lentiviral injection was performed starting from the third week of dietary regimen (SD or HFD). Each animal received 1 injection per week (at the third, fourth, and fifth week) with a total of three injections. The animals were left in their home cage, and an infusion cannula connected to a microsyringe (10 μL, Hamilton) by a polyethylene tube was inserted in the guide cannula. A total volume of 1 μL per hippocampus was injected in both hippocampi at a flow rate of 0.25 μL min$^{-1}$. The infusion cannula was left in place for an additional 2 min at the end of the infusion.

**Immunohistochemistry**. Mice were deeply anesthetized with ketamine and xylazine, and were transcardially perfused with PBS (0.1 M, pH 7.4) followed by 4% paraformaldehyde (PFA). Brains were collected, post-fixed overnight at 4 °C in PFA, and then transferred to a solution of 30% sucrose in 0.1 M PBS. Coronal section (36 μm) were then obtained using a cryostat (SLEE, Mainz, Germany) and subsequently stored at 4 °C in PBS until use. For Nissl staining the brains were

removed, fixed in a 4% PFA solution, frozen, sliced into 35 μm coronal sections, mounted on slides, and stained with Cresyl violet. The needle placements were verified with reference to the neuroanatomical Paxinos and Franklin mouse brain atlas[59]. Image from Mouse Brain Atlas is shown in Supplementary Fig. 4 according to the Allen brain atlas citation policy (http://www.alleninstitute.org/legal/citation-policy/). Data from mice with improper cannula placements (<15%) were excluded from the analyses.

**Statistical analysis**. Sample sizes were chosen with adequate power (0.8) according to results of prior pilot datasets or studies, including our own, that used similar methods or paradigms. Sample estimation and statistical analysis were performed using SigmaPlot 12 software. Data were first tested for equal variance and normality (Shapiro–Wilk test) and the appropriate statistical tests were chosen. The statistical tests employed in the experiments (i.e., Student's $t$-test, Mann–Whitney test, one-way ANOVA, and two-way ANOVA) are stated in the corresponding figure legends. We used Tukey (for LTP experiments), Student–Newman–Keuls (for recordings on autaptic cultures) or Bonferroni correction (for behavioral and molecular biology experiments, and LTP experiments in organotypic slices) for post hoc multiple analyses. All statistical tests were two-tailed and the level of significance was set at 0.05. Results are shown as mean ± SEM.

**Data availability**. The datasets generated and/or analyzed during the current study are available from the corresponding author on reasonable request.

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

## Acknowledgements

We thank Prof. Richard L. Huganir for DNA constructs. This work was supported by: intramural grants from Catholic University (Linea D.3.2-2013, D.3.2-2015, and Linea D.1 to C.G., Linea D.1 to S.F.); the Italian Ministry of University and Research (SIR 2014 RBSI14ZV59 to S.F.); the Italian Ministry of Health (GR-2011-02352187 to S.F.); Campania regional government (POR CAMPANIA FESR 2007/2013 O.O.2.1.-CUP B46D14002660009 to A.G.). We thank Patrizia Iannece for technical support.

## Author contributions

S.F. and C.G. conceived the study and supervised the work. M.S. and S.F. designed and performed molecular biology experiments. M.D., M.M., F.S., A.M., and C.R. performed electrophysiological experiments. D.D.L. and F.N. performed immunocytochemical experiments and confocal microscopy analyses. F.N. contributed to the western blotting experiments. R.L. and A.G. performed Gas Chromatography experiments. M.R., M.S., and S.F. carried out behavioral studies. S.F. and C.G. wrote the paper and all authors commented on the manuscript and approved its final version.

## Additional information

**Competing interests:** The authors declare no competing financial interests.

