## [Peer Review File · Nature Communications]

Reviewers' comments:

Reviewer #1 (Remarks to the Author):

The manuscript by Spinelli et al et al describes a novel link between high fat diet and changes in the posttranslational palmitoylation of synaptic proteins, synaptic plasticity and behavior. It is an incredibly interesting, well-executed, and in-depth study that will be of interest to a wide audience. Statistical analyses are appropriate. I only have minor comments.

1. Unless there is a clear reason for it, it would be preferable to use the standard 'zDHHHC3' nomenclature as opposed to the older 'GODZ' nomenclature.

2. Lievens (2016) demonstrated that the phosphorylation of zDHHHC3 significantly impacts its autopalmitoylation and ability to palm downstream substrates. I suggest a simple experiment to determine whether HFD (or IPA treatment in hippocampal cultures) impacts DHHHC3 autopalmitoylation and function.

3. The localization of PSD-95 at synapses also depends on palmitoylation and DHHHC3 is one of the PATs that palmitoylate PSD-95. Does IPA treatment impact the localization of PSD-95 at synapses? Is PSD-95 colocalized with other postsynaptic proteins following IPA treatment? This answer may alter the interpretation of the GluA1/PSD-95 colocalization assay (Fig 4E).

4. 2BP is a non-specific blocker of palmitoylation. Authors should comment on why they do not see changes in LTP induction in the presence of 2BP as palmitoylation of a number of substrates (e.g. PSD-95, delta-catenin) have been shown to be important for the establishment of LTP.

Reviewer #2 (Remarks to the Author):

This study showed that high fat diet (HFD) impairs hippocampal synaptic plasticity, learning and memory. The authors found that HFD increases expression of ZDHHHC3/GODZ palmitoyl-transferase and the GluA1 palmitoylation level, which reduces its cell-surface expression. In addition, they found that knockdown of ZDHHHC3/GODZ in the hippocampus or intranasal injection of 2-bromopalmitate (2-BP), an inhibitor of palmitoyl-transferases, reverts HFD- or IPA (insulin + palmitic acid treatment)-induced GluA1 hyper-palmitoylation and rescues impaired synaptic plasticity and hippocampus-dependent memory impairment in HFD mice. This work addresses an interesting issue how HFD affects cognitive functions in mice and the authors proposed the molecular mechanism through GluA1 palmitoylation. However, I feel that the data shown here is not sufficient to draw their conclusion. Rather, several data (Figs. 5 and 6) does not support their hypothesis due to the low specificity of experiments. Addressing the following points would improve and strengthen this paper.

Major comments

(1) Their hypothesis that HFD impairs hippocampal synaptic plasticity through affecting AMPAR palmitoylation might be reasonable given the previous report that ZDHHC3/GODZ overexpression traps AMPARs in the Golgi apparatus (Hayashi et al, Neuron 2005). To increase the specificity of their data, non-biased screening experiments, such as DNA microarray and palmitome using HFD mice, are necessary (Kang R et al, nature 2008).

(2) In Fig. 5, the authors performed the knockdown of ZDHHC3/GODZ in the hippocampus and showed reduced GluA1 palmitoylation and improvement of impaired memory in HFD mice. But, knockdown of ZDHHC3/GODZ affects the palmitoylation state of various substrates such as GABAA receptor gamma2 subunit (Fang et al, J Neurosci 2006), PSD-95 and so on. To directly show their hypothesis, the molecular replacement approach with palmitoylation-deficient GluA1 (or knock-in mouse approach) is necessary.

(3) In Fig. 6, the experiment using 2-BP does never support their conclusion. Because 2-BP treatment inhibits most of palmitoyl-transferase and reduces the palmitoylation levels of many substrates (e.g., GRIP, PICK and PSD-95). In fact, it was reported that treatment of hippocampal neurons with 2-BP reduces PSD-95 palmitoylation and AMPAR currents (El-Husseini et al, Cell 2002), which is an opposite result to the authors'.

(4) Data quality and quantification

I am concerned about the validity of data quantification. For example, it seems that the band intensities of Figs. 2a, 5f, and 6b do not reflect the values of corresponding graphs. Duplicated bands in Fig. 6g are too variable.

Minor comments

(1) In Fig. 4c, does AMPAR accumulate in the Golgi in IPA treated neurons? In addition, quantification and the statistical analysis in Fig. 4c are necessary.

(2) Line 65, "DHCC domain" should be "DHHC domain".

(3) Unification of Zdhhc3 (line 36) and GODZ (line136) is necessary throughout the manuscript.

(4) In Fig. 3e and 3f, the positions of "3, 6, 24 hour" are not correct.

(5) In supplementary Fig. 2, line 47, "almitoylation" should be "palmitoylation".

(6) Nomenclature, "GluA" is now more commonly used than "GluR".

(7) Method, line 217 (ABE method), I am wondering if the lysis buffer used here (1% NP-40, 1% Triton X-100) is sufficient for the extraction of all the palmitoylated proteins. The authors at least should confirm the extraction efficacy for AMPAR and NMDAR.

Reviewer #3 (Remarks to the Author):

Dear Editor,

It's known that metabolic derangements caused by HFD feeding are correlated with memory deficits, however, the exact molecular mechanisms are still elusive. In the manuscript, entitled "Brain insulin resistance impairs hippocampal synaptic plasticity and memory via

FoxO3a/Zdhhc3-dependent enhancement of GluR1 palmitoylation", Spinelli M et al elucidated a potential mechanism to bridge HFD-induced insulin resistance with impaired hippocampal synaptic plasticity and memory.

Here are the major findings:

1 HFD impairs synaptic plasticity including reduced EPSP amplitude and LTP. HFD increased fatty acids deposition in the hippocampus, including palmitic acid (PA). HFD also causes hyperinsulinemia, increased the basal phosphorylation of key insulin signaling molecules (eg Akt, GSK3beta, FoxO3a), yet abolished insulin induced phosphorylation of these molecules. 2 HFD increased the GluR1 palmitoylation, but reduced GluR1 phosphorylation (Ser845). Zdhhc3 (GODZ), a major palmitoyltransferase, was detected to have increased expression in HFD condition.

3 The authors also utilized in vitro system to show high insulin plus PA (IPA) caused insulin resistance and increased GluR1 palmitoylation. Godz transcription is regulated by FoxO3a, demonstrated by ChIP assays. The authors further showed IPA causes decreased AMPA response at the post-synaptic level (ie reduced mEPSC amplitude) and decreased GluR1 cell surface presentation.

4 Finally, the authors showed reducing GluR1 palmitoylation can rescue the HFD-induced impairment of memory and synaptic plasticity. Two independent mechanisms were used in the study, including knocking down Godz by shRNA or palmitoyl transferase inhibitor, 2-BP. Behavior data and LTP recordings both support the claim.

Overall, the topic is interesting and relevant. The authors used appropriate methods and made justified conclusion. The manuscript can be further improved by revising. Here are the reviewers major and minor critiques.

Major:

1. Figure 5f: the western blot (IB: streptavidin) can be shown with a more representative blot.
2. Figure 5a: the left panel showing EPSC amplitude is confusing. shCTR IPA and shGODZ VEH traces and legends are not matched.
3. Figure 6b,c,d,e,f,g: the experiment design can be improved by adding the CTR + 2-BP group.
4. 2-BP is not a specific inhibitor. Its on-target side effects / toxicity should be addressed.
5. Reduced GluR1 palmitoylation can ameliorate HFD-induced impairment of memory and synaptic plasticity. The key experiments were performed on brain slices. The hypothesis can be strengthened using transgenic animal models.

Minor:

1. Line 65 "DHCC"
2. Line 164 "aspecifically"

POINT-BY-POINT RESPONSE TO THE REFEREES' COMMENTS

Response to Reviewer #1

The manuscript by Spinelli et al. describes a novel link between high fat diet and changes in the posttranslational palmitoylation of synaptic proteins, synaptic plasticity and behavior. It is an incredibly interesting, well-executed, and in-depth study that will be of interest to a wide audience. Statistical analyses are appropriate. I only have minor comments.

>> (Response) **We are very grateful to the referee for his/her comments and suggestions that were very useful to improve our manuscript.**

1. Unless there is a clear reason for it, it would be preferable to use the standard 'zDHHC3' nomenclature as opposed to the older 'GODZ' nomenclature.

>> (Response) **The nomenclature 'GODZ' has been replaced with 'zDHHC3' throughout the text.**

2. Lievens (2016) demonstrated that the phosphorylation of zDHHC3 significantly impacts its autopalmitylation and ability to palm downstream substrates. I suggest a simple experiment to determine whether HFD (or IPA treatment in hippocampal cultures) impacts DHHC3 autopalmitylation and function.

>> (Response) **We followed the interesting suggestion of the reviewer and investigated the palmitoylation of zDHHC3 in the hippocampus of HFD mice. Our findings indicate that HFD increased the levels of palmitoylated zDHHC3 (Fig. 2f) along with the palmitoylation of some of its targets (e.g. GluA1, GluA2, GabaA Ry2; Supplementary Fig. 2b,c). Therefore, our findings suggest that HFD impacts on zDHHC3 through a dual mechanism: i) increasing the expression of PAT and ii) enhancing its activity by post-translational modifications.**

3. The localization of PSD-95 at synapses also depends on palmitoylation and DHHC3 is one of the PATs that palmitoylate PSD-95. Does IPA treatment impact the localization of PSD-95 at synapses? Is PSD-95 colocalized with other postsynaptic proteins following IPA treatment? This answer may alter the interpretation of the GluA1/PSD-95 colocalization assay (Fig 4E).

>> (Response) **As suggested by all reviewers, we analyzed the palmitoylation of a greater number of synaptic proteins, including PSD95. Actually, we found no significant alterations of PSD95 palmitoylation in the hippocampus of HFD-fed mice (Supplementary Fig. 2b). PSD95 palmitoylation was also unaffected in hippocampal neurons treated with IPA (Supplementary Fig. 3f). Finally, we evaluated the localization of PSD-95 at synapses by measuring the number of PSD95 puncta in dendrites and documented no significant changes upon IPA stimulation (Supplementary Fig. 3e). These findings support our interpretation that the reduction of GluA1/PSD95 co-localization at the synapse upon IPA treatment is primarily due to inhibition of GluA1 trafficking to the synapse.**

4. 2BP is a non-specific blocker of palmitoylation. Authors should comment on why they do not see changes in LTP induction in the presence of 2BP as palmitoylation of a number of substrates (e.g. PSD-95, delta-catenin) have been shown to be important for the establishment of LTP.

>> (Response) **2-BP may indeed act on numerous targets and modulate the palmitoylation of several synaptic proteins, leading to changes that may either promote (in case they occur on GluA1 or GluA2) or inhibit synaptic plasticity (in case they occur on PSD95 or GABAA receptor γ2). The scenario is also more complex because palmitoylation of different cysteine residues in the same target may induce opposite effects (e.g. NR2A or NR2B; Hayashi et al., Neuron 2009) and there is evidence that 2-BP may also inhibit depalmitoylating enzymes such as thioesterases (Pedro MP et al., PLoS One 2013). Consequently, it is very difficult to predict the net result of many, and potentially conflicting, effects exerted by 2-BP on different targets involved in the establishment of LTP.**

**To simplify the conveyed message, in our previous submission we did not include data we had, showing that LTP at hippocampal CA3-CA1 synapses was unaffected by 2-BP treatment in SD mice. These data have been added to the revised Fig. 6a and they are in agreement with the results of behavioral tests shown in Fig. 6b-d. Collectively, these findings suggest that, although 2-BP has the potential to affect multiple targets involved in synaptic plasticity, its net result, at the dose, time and method of administration we used, is no significant changes of LTP. Notably, the drug concentrations we used (5 µM and 0.25 nMol for *in vitro* and *in vivo* experiments, respectively) are lower than those used in other studies (≥ 20 µM; Dong et al., J Neurosci 2004; Li et al., J Biol Chem 2012; Davda D et al., ACS Chem Biol 2013).**

**However, in the revised manuscript we stressed that 2-BP is a "non-specific inhibitor of palmitoylation" (see page 10, line 247).**

∞∞∞∞∞∞∞∞∞∞∞∞∞∞∞∞∞∞∞∞∞∞∞∞∞∞∞∞∞∞∞∞∞∞∞∞∞∞∞∞∞∞∞∞∞∞∞∞∞∞∞∞

∞∞∞∞∞∞∞∞∞∞∞∞∞∞∞∞∞∞∞∞∞∞∞∞∞∞∞∞∞∞∞∞∞∞∞∞∞∞∞∞∞∞∞∞∞∞∞∞∞∞∞∞

**Response to Reviewer #2**

This study showed that high fat diet (HFD) impairs hippocampal synaptic plasticity, learning and memory. The authors found that HFD increases expression of ZDHHC3/GODZ palmitoyl-transferase and the GluA1 palmitoylation level, which reduces its cell-surface expression. In addition, they found that knockdown of ZDHHC3/GODZ in the hippocampus or intranasal injection of 2-bromopalmitate (2-BP), an inhibitor of palmitoyl-transferases, reverts HFD- or IPA (insulin + palmitic acid treatment)-induced GluA1 hyper-palmitoylation and rescues impaired synaptic plasticity and hippocampus-dependent memory impairment in HFD mice. This work addresses an interesting issue how HFD affects cognitive functions in mice and the authors proposed the molecular mechanism through GluA1 palmitoylation. However, I feel that the data shown here is not sufficient to draw their conclusion. Rather, several data (Figs. 5 and 6) does not support their hypothesis due to the low specificity of experiments.

Addressing the following points would improve and strengthen this paper.

Major comments

(1) Their hypothesis that HFD impairs hippocampal synaptic plasticity through affecting AMPAR palmitoylation might be reasonable given the previous report that ZDHHC3/GODZ overexpression traps AMPARs in the Golgi apparatus (Hayashi et al, Neuron 2005). To increase the specificity of their data, non-biased screening experiments, such as DNA microarray and palmitome using HFD mice, are necessary (Kang R et al, nature 2008).

>> (Response) **We agree that a broader spectrum analysis of gene expression and protein palmitoylation may provide critical information on the complexity of alterations occurring in the hippocampus of HFD-fed mice. However, systematic analyses of gene expression profile and palmitome in HFD mice, which are indeed in our highest interest, are beyond the scope of the present paper and they should not be dealt as a side information. On our opinion, these highly demanding studies have the dignity to be published as an independent follow-up paper (e.g Kang et al., Nature 2008; Dowal et al., Blood 2011; Marin et al., Cir Res 2012; Edmonds et al., Sci Rep 2017).**

To address the reviewer's concern, we have expanded the gene expression analysis of PATs and the study of synaptic proteins palmitoylation in the hippocampus of HFD mice. Specifically, we analyzed the expression of eleven genes coding for palmitoyl transferases targeting proteins involved in synaptic plasticity (Globo AK and Bamji SX, Current Opinion in Neurobiology 2017). The results of these experiments are shown in the revised Fig. 2e.

Moreover, we expanded the analysis of synaptic proteins by studying the palmitoylation of: i) all AMPA receptor subunits, ii) kainate and GABA receptor subunits known to be palmitoylated, and iii) PSD95. The results are shown in Supplementary Fig. 2b.

(2) In Fig. 5, the authors performed the knockdown of ZDHHC3/GODZ in the hippocampus and showed reduced GluA1 palmitoylation and improvement of impaired memory in HFD mice. But, knockdown of ZDHHC3/GODZ affects the palmitoylation state of various substrates such as GABAA receptor gamma2 subunit (Fang et al, J Neurosci 2006), PSD-95 and so on. To directly show their hypothesis, the molecular replacement approach with palmitoylation-deficient GluA1 (or knock-in mouse approach) is necessary.

>> (Response) **We thank the reviewer for this very useful comment that prompted us to perform new experiments whose results strengthened our conclusions. Specifically, we biolistically transfected organotypic slices with GluA1 wt or palmitoylation-deficient GluA1 (GluA1 C585S/C811S) and studied the effects of IPA on LTP. While IPA still impaired LTP at hippocampal CA3-CA1 synapses in organotypic slices overexpressing GluA1 wt, the transfection of palmitoylation-deficient GluA1 completely abolished the IPA detrimental effects (Fig. 5g).**

Although HFD also affects the palmitoylation of other ZDHHC3 targets in the hippocampus (e.g. GABAA receptor gamma2 subunit, as shown in Supplementary Fig. 2b), we believe that these findings clearly demonstrate the critical role of GluA1 hyper-palmitoylation in the insulin resistance-dependent impairment of synaptic plasticity.

(3) In Fig. 6, the experiment using 2-BP does not support their conclusion. Because 2-BP treatment inhibits most of palmitoyl-transferase and reduces the palmitoylation levels of many substrates (e.g., GRIP, PICK and PSD-95). In fact, it was reported that treatment of hippocampal neurons with 2-BP reduces PSD-95 palmitoylation and AMPAR currents (El-Husseini et al, Cell 2002), which is an opposite result to the authors'.

>> (Response) **First, we would like to draw the reviewer's attention on the 2-BP concentrations used in our experiments (5 μ M and 0.25 nMol for *in vitro* and *in vivo* experiments, respectively) that are much lower than that used by El-Husseini et al. (20 μ M). The differences observed in the effect of 2-BP might be also due to the different experimental models (i.e., 24 hours-stimulation in organotypic slices medium or chronic intranasal injection in mice versus 8 hour-treatment of hippocampal neuronal culture).**

To further address this issue, we measured spontaneous miniature excitatory postsynaptic current and AMPA-mediated current density in hippocampal organotypic slices treated with 5 μ M 2-BP and found no significant changes in any of the studied parameters. Specifically, mEPSC frequency was 0.71 ± 0.17 (n=14) and 0.71 ± 0.07 Hz (n=17) in vehicle- and 2-BP-treated slices, respectively ($p = 0.99$); mEPSC

amplitude was 19.1 ± 2.4 and 15.6 ± 1.8 pA, respectively ($p = 0.26$); AMPA current density was 8.8 ± 0.4 and 10.1 ± 0.3 pA μF^{-1} ($p = 0.48$); statistics by unpaired Student's t-test.

As already mentioned in the response to the Reviewer #1 (point 4), 2-BP may have multifaceted effects on numerous targets and modulate the palmitoylation of several synaptic proteins, inducing changes either promoting (in case they occur on GluA1 or GluA2) or impinging on synaptic plasticity (in case they occur on PSD95 or GABAA receptor $\gamma 2$). Moreover, there are potential targets whose effect depends on the palmitoylated cysteine residue (e.g. NR2A or NR2B; Hayashi et al., Neuron 2009) and there is evidence that 2-BP may also inhibit depalmitoylating enzymes such as thioesterases (Pedro MP et al., PLoS One 2013).

To confirm our *ex vivo* data we also performed the 2-BP administration *in vivo*. Data showing that 2-BP alone does not affect LTP at CA3-CA1 synapses have been added to the revised Fig. 6a.

(4) Data quality and quantification

I am concerned about the validity of data quantification. For example, it seems that the band intensities of Figs. 2a, 5f, and 6b do not reflect the values of corresponding graphs. Duplicated bands in Fig. 6g are too variable.

>> (Response) **As requested by the reviewer we replaced data shown in Figs. 2a (GluA1 palmitoylation in HFD) and 5f (GluA1 palmitoylation in LV-shzDHHC3) with more representative blots. The previous Fig. 6g (now Fig. 6e) has been changed to include the effects of 2-BP treatment on GluA1 palmitoylation in SD mice. Fig. 6b has been moved to Supplementary Fig. 5b.**

Minor comments

(1) In Fig. 4c, does AMPAR accumulate in the Golgi in IPA treated neurons? In addition, quantification and the statistical analysis in Fig. 4c are necessary.

>> (Response) **GluA1 immunofluorescence in the Golgi apparatus has been analyzed confirming an increase of GluA1 localization in the Golgi upon IPA treatment. The results of these new experiments have been added to Supplementary Fig. 3d. Data shown in Fig 4c have been quantified and results are shown in Supplementary Fig. 3c.**

(2) Line 65, "DHCC domain" should be "DHHC domain".

>> (Response) **The text has been corrected.**

(3) Unification of Zdhhc3 (line 36) and GODZ (line 136) is necessary throughout the manuscript.

>> (Response) **We replaced the nomenclature 'GODZ' with 'zDHHC3' throughout the manuscript.**

(4) In Fig. 3e and 3f, the positions of "3, 6, 24 hour" are not correct.

>> (Response) **We apologize for the inaccuracy. The figure labeling has been corrected.**

Overall, the topic is interesting and relevant. The authors used appropriate methods and made justified conclusion. The manuscript can be further improved by revising. Here are the reviewers major and minor critiques.

>> (Response) **We wish to thank the referee for the kind words of appreciation towards our work.**

Major:

1. Figure 5f: the western blot (IB: streptavidin) can be shown with a more representative blot.

>> (Response) **The blot shown in the previous Fig. 5f (now Fig. 5e) has been changed.**

2. Figure 5a: the left panel showing EPSC amplitude is confusing. shCTR IPA and shGODZ VEH traces and legends are not matched.

>> (Response) **The legend of Fig. 5a has been corrected.**

3. Figure 6b,c,d,e,f,g: the experiment design can be improved by adding the CTR + 2-BP group.

>> (Response) **As suggested by the reviewer we integrated figure 6 with CTR + 2-BP group (now named SD_{2BP}). We had already performed LTP experiments on mice fed with standard diet and injected with 2-BP and the results were in line with those obtained from organotypic slices. These data have been now added to Fig. 6a. Moreover, we carried out a novel battery of behavioral tests and new molecular experiments with all four experimental groups (SD_{VEH}, SD_{2BP}, HFD_{VEH} and HFD_{2BP}). The results of these experiments are shown in Fig. 6b-e.**

4. 2-BP is not a specific inhibitor. Its on-target side effects / toxicity should be addressed.

>> (Response) **We have investigated the toxicity of 2-BP on primary cultures of hippocampal neurons. Our data indicating no toxic effects of the palmitoylation inhibitor, at doses and times used in our experimental protocols, are shown in Supplementary Figure 5c.**

5. Reduced GluR1 palmitoylation can ameliorate HFD-induced impairment of memory and synaptic plasticity. The key experiments were performed on brain slices. The hypothesis can be strengthened using transgenic animal models.

>> (Response). **We agree with the reviewer that investigating the impact of HFD in a mouse model expressing palmitoylation-deficient GluA1 mutant would strengthen our conclusion. To test our hypothesis that GluA1 hyper-palmitoylation is the critical determinant of the observed effects we took advantage of biolistic transfection of plasmid encoding palmitoylation-deficient GluA1 in organotypic slices. We found that transfection of the palmitoylation-deficient GluA1 mutant abolished the detrimental effects of IPA on LTP (Fig. 5g).**

Minor:

1. Line 65 "DHCC"

2. Line 164 “aspecifically”

>> (Response) **Text has been corrected.**

REVIEWERS' COMMENTS:

Reviewer #1 (Remarks to the Author):

The manuscript by Spinelli et al et al describes a novel link between high fat diet and changes in the posttranslational palmitoylation of synaptic proteins, synaptic plasticity and behavior. The original manuscript was well-executed and I had made a few suggestions for additional experiments that would address the interpretation of some of the data. The revised manuscript has additional data that help clarify some of the conclusions of the study and continues to be a strong study.

Although I continue to believe that that 2BP is a very messy compound to use (as most articulately expressed by the authors in their rebuttal), it is worth including the 2BP data. I would, however, suggest that the authors include the caveats in the discussion. They have added, "Actually, we cannot rule out that the effects of HFD on hippocampal synaptic plasticity may partly depend on altered palmitoylation of other zDHHC3 targets, as suggested by the increased palmitoylation of GabaA Ry2 (Supplementary Fig. 2b,c)", but the lack of specificity goes beyond that and it should be well-noted in the discussion.

Reviewer #2 (Remarks to the Author):

The authors have now performed several experiments to address reviewers' concerns. Although I suggested the molecular replacement experiment with palmitoylation-deficient GluA1 (that means knockdown of GluA1 and re-expression of knockdown-resistant GluA1 mutant) or knock-in mouse approach, the authors just examined the effect of overexpression of palmitoylation-deficient GluA1 (Fig. 5g). This experiment is not sufficient to support their conclusion.

Although it was reported that treatment of hippocampal neurons with 2-BP reduces PSD-95 palmitoylation and AMPAR currents (El-Husseini et al, Cell 2002), the authors claimed that treatment with 2-BP (5 micro M) alone did not affect the basal synaptic transmission and the LTP amplitude under their conditions. They assumed that the different results might be caused by the different experimental conditions. Even if this is the case, they did not neglect off-target effects of 2-BP. In fact, they described that 2-BP is the non-specific inhibitor of palmitoylation (line 248). Therefore, the data presented here does not strengthen their conclusion even if 2-BP treatment consistently reverted GluA1 palmitoylation to the control level (Supplementary Fig. 5b). Again, the best experiment is knock-in mouse approach (palmitoylation-deficient GluA1).

Reviewer #3 (Remarks to the Author):

Dr. Fusco and colleagues revised the manuscript entitled "Brain insulin resistance impairs

hippocampal synaptic plasticity and memory via FoxO3a/zDHH3-dependent enhancement of GluA1 palmitoylation".

The authors addressed the reviewer's comments by and large.

"4. 2-BP is not a specific inhibitor. Its on-target side effects / toxicity should be addressed."

"

The authors addressed the "toxicity" issue without commenting on the "on-target side effects". The authors elaborated in more details to reviewer #1 point #4 and reviewer #2 point #3, which are similar questions.

"5. Reduced GluR1 palmitoylation can ameliorate HFD-induced impairment of memory and synaptic plasticity. The key experiments were performed on brain slices. The hypothesis can be strengthened using transgenic animal models. "

The authors generated a "GluA1 mutant" for this comment. However, the reviewer was asking about generating a KO mouse based on Figure 5 and assaying the phenotypes subsequently. The reviewer understands the time and effort needed for doing this experiment. The authors can make a point in the discussion in addition to the existing content (line 331-333).

POINT-BY-POINT RESPONSE TO THE REFEREES' COMMENTS

Reviewer #1 (Remarks to the Author):

The manuscript by Spinelli et al et al describes a novel link between high fat diet and changes in the posttranslational palmitoylation of synaptic proteins, synaptic plasticity and behavior. The original manuscript was well-executed and I had made a few suggestions for additional experiments that would address the interpretation of some of the data. The revised manuscript has additional data that help clarify some of the conclusions of the study and continues to be a strong study.

Although I continue to believe that that 2BP is a very messy compound to use (as most articulately expressed by the authors in their rebuttal), it is worth including the 2BP data. I would, however, suggest that the authors include the caveats in the discussion. They have added, "Actually, we cannot rule out that the effects of HFD on hippocampal synaptic plasticity may partly depend on altered palmitoylation of other zDHHC3 targets, as suggested by the increased palmitoylation of GabaA R γ 2 (Supplementary Fig. 2b,c)", but the lack of specificity goes beyond that and it should be well-noted in the discussion.

>> (Response) As suggested by all the reviewers and the Editor, we further stressed the non-specificity of the palmitoylation inhibitor 2-BP and included caveats in the discussion (lines 361-371). We would also like to stress that the causal link between GluA1 palmitoylation and cognitive impairment in our brain insulin-resistance model does not rely on the results of 2-BP experiments that were primarily aimed at paving the way to future clinical studies. Now, this is better indicated in the section "Results" (lines 267-270).

Reviewer #2 (Remarks to the Author):

The authors have now performed several experiments to address reviewers' concerns. Although I suggested the molecular replacement experiment with palmitoylation-deficient GluA1 (that means knockdown of GluA1 and re-expression of knockdown-resistant GluA1 mutant) or knock-in mouse approach, the authors just examined the effect of overexpression of palmitoylation-deficient GluA1 (Fig. 5g). This experiment is not sufficient to support their conclusion.

>> (Response) We agree with the Reviewer that the knock-in mouse approach would strengthen our conclusions, and we definitely plan to address this issue in our future studies, as stated in the Discussion (lines 353-354). With regard to our *ex vivo* experiments, we preferred to overexpress the palmitoylation-deficient GluA1 alone rather than combining the silencing of naive GluA1 with the overexpression of the mutant GluA1 in organotypic slices because the latter approach may give rise to complex scenarios depending on whether the two plasmids are differently transfected. Indeed, we might have neurons with only lower expression of naive GluA1, neurons with naive GluA1 replaced by the palmitoylation-deficient GluA1 and neurons overexpressing mutant GluA1 together with naive GluA1. This would introduce a great variability in the studied responses, thus undermining the result reliability.

Although it was reported that treatment of hippocampal neurons with 2-BP reduces PSD-95 palmitoylation and AMPAR currents (El-Husseini et al, Cell 2002), the authors claimed that treatment with 2-BP (5 micro M) alone did not affect the basal synaptic transmission and the LTP amplitude under their conditions. They assumed that the different results might be caused by the different experimental conditions. Even if this is the case, they did not neglect off-target effects of 2-BP. In fact, they described that 2-BP is the non-specific inhibitor of palmitoylation (line 248). Therefore, the data presented here does not strengthen their conclusion even if 2-BP treatment consistently reverted GluA1 palmitoylation to the control level (Supplementary Fig. 5b). Again, the best experiment is knock-in mouse approach (palmitoylation-deficient GluA1).

>> (Response) We fully agree with the reviewer that 2-BP experiments do not demonstrate the causal link between GluA1 palmitoylation and cognitive impairment in our brain insulin-resistance models. As clarified in the revised Results (lines 267-270) these experiments were performed to get preliminary information potentially useful for planning future clinical studies aimed at developing new pharmacological strategies against cognitive decline in metabolic diseases.

Reviewer #3 (Remarks to the Author):

Dr. Fusco and colleagues revised the manuscript entitled "Brain insulin resistance impairs hippocampal synaptic plasticity and memory via FoxO3a/zDHHHC3-dependent enhancement of GluA1 palmitoylation".

The authors addressed the reviewer's comments by and large.

"4. 2-BP is not a specific inhibitor. Its on-target side effects / toxicity should be addressed. "

The authors addressed the "toxicity" issue without commenting on the "on-target side effects". The authors elaborated in more details to reviewer #1 point #4 and reviewer #2 point #3, which are similar questions.

>> (Response) We implemented the discussion according to the Reviewer's request (lines 361-371).

"5. Reduced GluR1 palmitoylation can ameliorate HFD-induced impairment of memory and synaptic plasticity. The key experiments were performed on brain slices. The hypothesis can be strengthened using transgenic animal models. "

The authors generated a "GluA1 mutant" for this comment. However, the reviewer was asking about generating a KO mouse based on Figure 5 and assaying the phenotypes subsequently. The reviewer understands the time and effort needed for doing this experiment. The authors can make a point in the discussion in addition to the existing content (line 331-333).

>> (Response) The authors integrated the discussion according to the Reviewer's suggestion (lines 353-354).